

# High-resolution spatial patterns and drivers of terrestrial ecosystem carbon dioxide, methane, and nitrous oxide fluxes in the tundra

Anna-Maria Virkkala[1,2], Pekka Niittynen[3], Julia Kemppinen[4], Maija E. Marushchak[5], Carolina Voigt[5], Geert Hensgens[6], Johanna Kerttula[5], Konsta Happonen[7], Vilna Tyystjärvi[8], Christina Biasi[5], Jenni Hultman[9,10], Janne Rinne[9], Miska Luoto[2]

[1]Woodwell Climate Research Center, Falmouth, 149 Woods Hole Road, MA, USA
[2]University of Helsinki, Department of Geosciences and Geography, Gustaf Hällströmin katu 2, 00014 University of Helsinki, Finland
[3]University of Jyväskylä, Department of Biological and Environmental Science, P.O. Box 35 FI-40014, Jyväskylä, Finland
[4]Geography Research Unit, University of Oulu, P.O. Box 8000 FI-90014, Oulu, Finland
[5]University of Eastern Finland, Department of Environmental and Biological Sciences, P.O. Box 1627 FI- 70211, Kuopio, Finland
[6]Vrije Universiteit Amsterdam, Department of Earth and Climate, De Boelelaan 1085, 1081 HV, Amsterdam, the Netherlands
[7]Youth Research Society, Kumpulantie 3 A, 00520 Helsinki, Finland
[8]Finnish Meteorological Institute, Climate System Research Unit, Erik Palménin aukio 1, FI-00560 Helsinki, Finland
[9]Natural Resources Institute Finland, Latokartanonkaari 9, 00790 Helsinki, Finland
[10]University of Helsinki, Department of Microbiology, Viikinkaari 9, 00014 University of Helsinki, Finland
*Correspondence to*: Anna-Maria Virkkala, avirkkala@woodwellclimate.org

**Abstract**. Arctic terrestrial greenhouse gas (GHG) fluxes of carbon dioxide ($CO_2$), methane ($CH_4$) and nitrous oxide ($N_2O$) play an important role in the global GHG budget. However, these GHG fluxes are rarely studied simultaneously, and our understanding of the conditions controlling them across spatial gradients is limited. Here, we explore the magnitudes and drivers of GHG fluxes across fine-scale terrestrial gradients during the peak growing season (July) in sub-Arctic Finland. We measured chamber-derived GHG fluxes and soil temperature, soil moisture, soil organic carbon and nitrogen stocks, soil pH, soil carbon-to-nitrogen (C/N) ratio, soil dissolved organic carbon content, vascular plant biomass, and vegetation type from 101 plots scattered across a heterogeneous tundra landscape (5 km$^2$). We used these field data together with high-resolution remote sensing data to develop machine learning models to predict (i.e., upscale) daytime GHG fluxes across the landscape at 2-m resolution. Our results show that this region was on average a daytime net GHG sink during the growing season. Although our results suggest that this sink was driven by $CO_2$ uptake, it also revealed small but widespread $CH_4$ uptake in upland vegetation types, shifting this region to an average net $CH_4$ sink at the landscape scale during growing season, despite the presence of high-emitting wetlands. Average $N_2O$ fluxes were negligible. $CO_2$ fluxes were controlled primarily by annual average soil temperature and biomass (both increase net sink) and vegetation type, $CH_4$ fluxes by soil moisture (increases net emissions) and vegetation type, and $N_2O$ fluxes by soil C/N (lower C/N increases net source). These results demonstrate the potential of high spatial resolution modelling of GHG fluxes in the Arctic. They also reveal the dominant role of $CO_2$ fluxes across the tundra landscape but suggest that $CH_4$ uptake might play a significant role in the regional GHG budget.



## 1 Introduction

Over the past millennia, Arctic soils in the treeless tundra biome have played an important role in the global climate system by accumulating large amounts of carbon (C) and nitrogen (N), thus cooling the climate (Hugelius et al., 2014, 2020; Strauss et al., 2017). However, the ongoing climate warming is changing the C and N cycles, leading to potentially increased net greenhouse gas (GHG) emissions from Arctic ecosystems to the atmosphere (Belshe et al., 2013; McGuire et al., 2012; Masyagina and Menyailo, 2020). Yet, even the current GHG balance of Arctic ecosystems is insufficiently understood due to severe gaps in flux measurement networks and poorly performing coarse-resolution models (Virkkala et al., 2021; Treat et al., 2018c). Thus, the contribution of Arctic regions to the global climate feedback remains uncertain.

One of the main uncertainties in understanding the Arctic GHG balance is related to the inadequately quantified magnitudes of all three main GHG fluxes - carbon dioxide ($CO_2$), methane ($CH_4$) and nitrous oxide ($N_2O$) - which show pronounced spatial variability across the diverse terrestrial environmental gradients in tundra (Virkkala et al., 2018; Pallandt et al., 2021; Voigt et al., 2020). In most tundra ecosystems, $CO_2$ fluxes are the largest flux driving the GHG balance due to the strong growing season photosynthetic activity and relatively high non-growing season respiratory $CO_2$ losses (Natali et al., 2019; Euskirchen et al., 2012; Heiskanen et al., 2021). However, growing evidence points to the importance of $CH_4$ and $N_2O$ fluxes, which are more potent GHGs than $CO_2$ (Voigt et al., 2017b). These two trace gases can have considerable variation between sink and source activity in the tundra, and they have different spatiotemporal dynamics with each other and compared to $CO_2$ fluxes (Emmerton et al., 2014; Bruhwiler et al., 2021). However, only a few studies have simultaneously considered the contributions of all three main GHG fluxes to the tundra GHG balance (Voigt et al., 2017b; Kelsey et al., 2016; Brummell et al., 2012; Wagner et al., 2019).

The largest fine-scale differences in Arctic GHG fluxes occur in ecosystems with spatially varying soil moisture conditions (McGuire et al., 2012). Broadly speaking, the Arctic can be divided into wetlands and drier uplands (i.e., shrublands, grasslands, and barren lands; see e.g. (Treat et al., 2018a; Virkkala et al., 2021). Wetlands cover between 5 and 25 % of the Arctic (Olefeldt et al., 2021; Kåresdotter et al., 2021; Raynolds et al., 2019). They are hotspots for soil C and N stocks and have the potential for high $CH_4$ emissions (Euskirchen et al., 2014; Hugelius et al., 2020); therefore they have been intensively studied (Rinne et al., 2018; Peltola et al., 2019; Turetsky et al., 2014). However, uplands cover the largest part of the Arctic (75 to 95 %) and can have significant variability in environmental conditions and GHG fluxes. These uplands have been relatively well studied for $CO_2$ fluxes (Williams et al., 2006; Cahoon et al., 2012a). Upland $CH_4$ and $N_2O$ fluxes, on the other hand, remain less well understood in terms of their magnitudes and drivers (Virkkala et al., 2018; Voigt et al., 2020). There are still likely some GHG flux hotspots to be discovered and coldspots to be verified, particularly in the upland tundra ecosystems.

The Arctic tundra is characterised by fine-scale environmental heterogeneity even within upland and wetland tundra environments. Thus, local-scale study settings that cover the main spatial environmental gradients are highly important (Treat et al., 2018c; Davidson et al., 2017). Such fine-scale variabilities are often measured with chambers, but most chamber-based study designs are limited to relatively small environmental gradients focusing on only a handful of different land cover types and environmental variables, leaving large gaps in our





understanding of GHG flux hotspots (Virkkala et al. 2018). In this study, using an extensive spatial study design
with chamber GHG flux measurements from 101 plots, we aim to understand the magnitudes and environmental
drivers of Arctic terrestrial $CO_2$, $CH_4$, and $N_2O$ fluxes in a heterogeneous tundra landscape dominated by upland
heaths. By combining in-situ measurements and remote sensing data, we investigate the fine-scale (2 m) spatial
heterogeneity of GHG fluxes across the landscape, and estimate the contribution of the three gases to the total
landscape-scale GHG flux.

## 2 Materials and Methods

### 2.1 Study area

The field measurements were collected during 2016-2018 in a sub-Arctic tundra environment in Kilpisjärvi
(Gilbbesjávri in Northern Sámi language), northwestern Finland (69.06 N, 20.81 E). The study area is located on
an elevational gradient between two fells, Saana (Sána; 1029 m.a.s.l) and Korkea-Jehkats (Jiehkkáš; 960 m.a.s.l),
and the valley in between (~600 m.a.s.l.). The study area is above the mountain birch (*Betula pubescens* ssp.
*czerepanovii*) forest and is dominated by dwarf-shrub evergreen and deciduous heaths. Dominant vascular plant
species are, e.g., *Empetrum nigrum* ssp. *hermaphroditum*, *Betula nana*, *Vaccinium myrtillus, Vaccinium vitis-*
*idaea,* and *Phyllodoce caerulea*. Vegetation in the wetlands is dominated by species common to fen wetlands,
such as *Eriophorum* sp. or *Carex* sp. Mesic meadows are rich in forbs and grasses whereas barren heaths
accommodate mostly lichens (e.g. *Cladonia* spp.) and mat-forming cushion plants (e.g. *Diapensia lapponica*) with
scattered patches of *E. nigrum* and *B. nana*. Soils in the area are shallow (mean organic layer depth 6.6 cm, mean
mineral layer depth 13.0 cm), and permafrost is absent from soils but can be found in the bedrock above 800 m
a.s.l. (King and Seppälä, 1987). The environment is relatively undisturbed but experiences reindeer (*Rangifer*
*tarandus tarandus*) grazing. The mean annual temperature in Saana fell (1002 m.a.s.l.) is -3.1 °C and the annual
precipitation in Kilpisjärvi village ca. 5 km from the study site (480 m.a.s.l.) is 518 mm in 1991-2018 (Finnish
Meteorological Institute, 2019a, b).
Our study design covered an area of ca. 3 x 1.5 km and consisted of 101 GHG flux measurement plots and 50 to
5280 plots with other environmental data (Fig. 1). We selected the plots based on a combination of stratified
sampling and systematic grid approaches, and the plots contain a variety of environmental gradients and habitats
as well as the transition zones between them (Kemppinen et al., 2021). We recorded the locations of the plots
using a hand-held Global Navigation Satellite System receiver with an accuracy of up to ≤6 cm under optimal
conditions (GeoExplorer GeoXH 6000 Series; Trimble Inc., Sunnyvale, CA, USA).

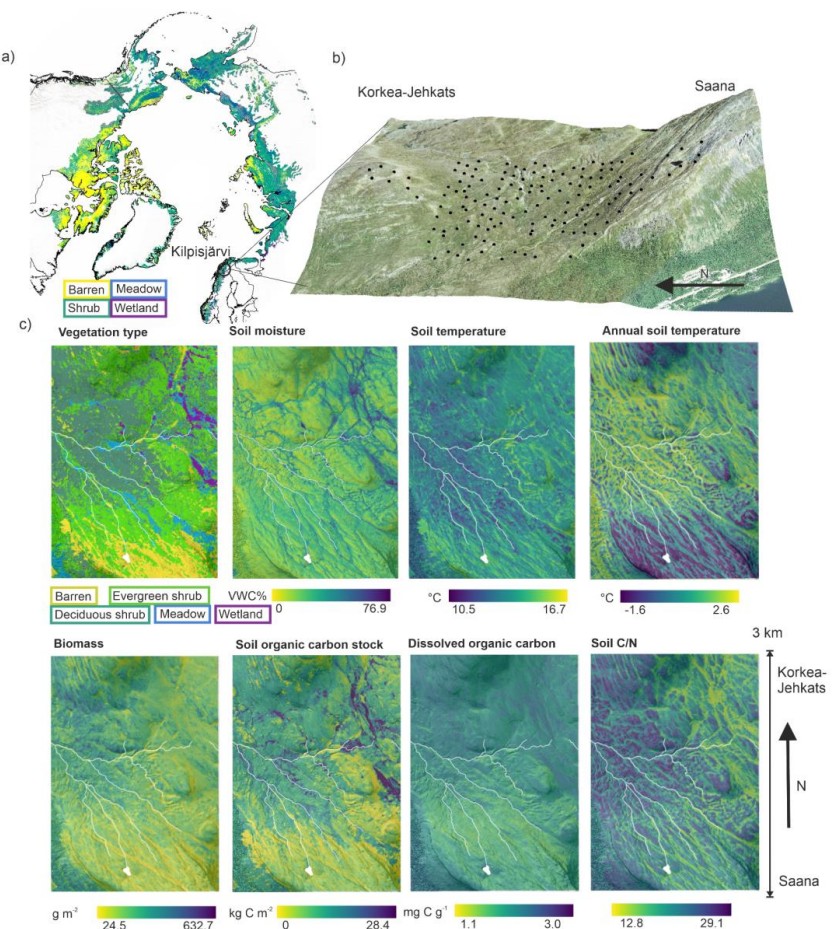


**Figure 1**: The distribution of the main vegetation types across the Arctic tundra (Dinerstein et al., 2017; Agency,
2017) and the location of our study area (a), the distribution of plots (b) and environmental conditions derived
from statistical upscaling of in-situ measurements (see Sect. 2.4.2 Machine learning models) across the study area
(c). Soil moisture and temperature represent mean daytime (8 am to 8 pm) conditions from the 1st of July to the
2nd of August and annual soil temperature is an average within the entire year (July 2017-June 2018). Other
conditions represent growing season conditions and are considered static in this study. The aerial image is
produced by the National Land Survey of Finland (accessed in 2016).

**2.2 Data**
We measured GHG fluxes from 101 plots during the peak growing season (from now on, growing season).
Environmental conditions explaining these GHG fluxes were measured from 73-100 % of these plots; missing
data were filled using the environmental predictions (see Sect. 2.4.2 Machine learning models, Table S1). We
used additional in-situ environmental data to upscale and visualize environmental conditions across the entire




landscape (see Sect. 2.4.2 Machine learning models and Fig. 2): continuous soil moisture loggers (50 plots),
continuous soil temperature loggers (250), soil samples for carbon and nitrogen stock and C/N estimation (168),
and vegetation classification data (5280). The full set of variables at a plot consisted of the plot for GHG flux
measurements, and of a nearby complementing plot (max. 2 m distance) where we monitored soil moisture and
temperature continuously and did a vegetation classification. The additional plot was separated from the GHG
plot to avoid disturbance of the continuous recordings. The additional plot was carefully situated to similar
vegetation and microtopographic conditions as the GHG plot. Soil samples were collected as close as possible to
the GHG plot.

**2.2.1 Chamber measurements**

We measured GHG exchange using a static, non-steady state non-flow-through system (Livingston and
Hutchingson, 1995) composed of an acrylic chamber (20 cm diameter, 25 cm height). The chamber was placed
on top of a collar and ventilated before each measurement. Prior to the measurements, we installed steel collars,
which were 21 cm in diameter and 6-7 cm in height. Each collar was visited once during the growing-season ,
and measurements were conducted between 10 am and 5 pm. Although we did not have any temporal replicates,
the spatial variation in our plots covered most of the temperature variation during the growing season. For more
details, see Sect. S1.
For $CO_2$ flux measurements, transparent and opaque chamber measurements were conducted during 1st of July
and 27th of July, 2018. The chamber included a small fan, a carbon dioxide probe GMP343 and an air humidity
and temperature probe HMP75 (Vaisala, Finland). In the chamber, $CO_2$ concentration, air temperature and
relative air humidity were recorded at 5-s intervals for 90 s. Photosynthetically active radiation was logged
manually outside the chamber at 10-s intervals during the same period using a MQ-200 quantum sensor with a
hand-held meter (Apogee Instruments, Inc, USA). MQ-200 measures photosynthetic photon flux density
(PPFD) at a spectral range from 410 to 655 nm in $\mu mol\ m^{-2}\ s^{-1}$. For more details of the equipment, see
Happonen et al. (2022).
We progressively decreased the light intensity of net ecosystem exchange (NEE) measurements from ambient
conditions to ca. 80%, 50% and 30% PPFD by shading the chamber with layers of white mosquito net (replicate
measurements per collar = 5 - 9). Ecosystem respiration (ER) was measured in dark conditions (0 PPFD), which
were obtained by covering the chamber with a space blanket (replicates = 2 - 3). Before flux calculations, we
discarded the first 0 - 5 s as well as the last 5 s of the measurements to remove potentially disturbed
observations. Fluxes were calculated from the concentration change within the chamber headspace over time
using linear regression (for performance statistics see Sect. S2).
We standardized NEE, GPP, and ER measurements conducted at different light and temperature conditions to
allow across-plot comparison of the fluxes. We fitted light-response curves using a non-linear hierarchical
bayesian model with the plot as a random effect (Sect. S5). We used the Michaelis-Menten equation to model
instantaneous NEE as a function of plot-specific ER, maximum photosynthetic rate (GPP$_{max}$) and the half-
saturation constant (K) using the same formula as in (Williams et al., 2006; Cahoon et al., 2012b). ER also had
an exponential air temperature (T) response (for more details, see (Happonen et al., 2022). We used this model





to predict NEE at dark (0 PPFD, i.e. ER) and average light (600 PPFD) conditions, and an air temperature of
20°C at each plot. 20°C was chosen as it represents a typical air temperature inside the chamber during flux
measurements, and 600 PPFD because it is widely used in tundra literature (Dagg and Lafleur, 2011; Shaver et
al., 2007). We then subtracted ER from the NEE normalized to average light conditions to arrive at an estimate
of normalized gross primary productivity (GPP). Negative values in NEE indicate a net sink of $CO_2$ from the
atmosphere to the ecosystems. GPP and ER are given as positive values.
We measured $CH_4$ and $N_2O$ fluxes with an opaque chamber (0 PPFD). Measurements were conducted during
the 2nd of July and 2nd of August, 2018. Five gas samples were taken within a 50-min enclosure time and
transferred into 12-mL vials (Labco Exetainer, Labco Ltd.). The vials were pre-evacuated in the laboratory and
filled with 25 mL of the sample in the field. Gas samples were analyzed at the University of Eastern Finland
with a gas chromatograph (Agilent 7890B; Agilent Technologies, Santa Clara, CA, USA), equipped with an
autosampler (Gilson Inc., Middleton, WI, USA), with thermal conductivity detector (TCD) for $CO_2$, flame
ionization detector (FID) for $CH_4$ and an electron capture detector (ECD) for $N_2O$. We calculated gas
concentrations from GC peak areas relative to peak areas derived by analyzing gas standards ($CO_2$: 7
concentration levels ranging from 0-10000 ppm; $CH_4$: 7 concentration levels ranging from 0-100 ppm; $N_2O$: 5
concentration levels ranging from 0-5000 ppb). Fluxes were calculated from the concentration change within the
chamber headspace over time using linear regression. Quality control was based on visual inspection and
RMSE. We also verified that the RMSE was less than 3 * standard deviation of gas standards in a similar
concentration range. Negative values in these fluxes represent net $CH_4$ and $N_2O$ sinks from the atmosphere to
the ecosystems.
**2.2.2 Soil temperature and moisture data**
Soil moisture and soil temperature were measured simultaneously during the flux measurements. We measured
soil moisture with a time-domain reflectometry sensor (FieldScout TDR 300; Spectrum Technologies Inc.,
Plainfield, IL, USA; 0 to 7.5 cm depth). Soil temperature measurements conducted at the same time as $CO_2$ flux
measurements were taken with a thermometer (TD 11 thermometer; VWR International bvba; Leuven,
Germany; 6.0 to 7.5 cm depth). Soil temperature measurements (TM-80N measure and ATT-50 sensor)
conducted at the same time as $CH_4$ and $N_2O$ flux measurements were taken with a thermometer in the uppermost
10 cm. We refer to these variables as soil moisture and soil temperature throughout the text.
Temperature loggers (Thermochron iButton DS1921G and DS1922L, San Jose, CA, USA and TMS-4; TOMST
s.r.o., Prague, Czech Republic) monitored temperatures at 7.5 cm and 6.5 cm (iButton and TMS-4, respectively)
belowground at 0.25–4 h intervals (Sect. S3). We calculated a variable describing soil temperature conditions
during the previous 12 months by averaging the iButton measurements from the study design (n=138) from July,
2017 to June 2018. We refer to this variable as annual soil temperature. In addition to temperature, the TMS-4
loggers also monitored soil moisture (raw time-domain transmission data between 1 and 4095) to a depth of c.
14 cm (Wild et al., 2019). The raw time-domain transmission data was transformed into volumetric water
content (VWC%) (Tyystjärvi et al., 2022).



These continuous soil moisture and temperature data were used to upscale soil microclimatic conditions at 2-
hour timesteps during daytime (8 am to 8 pm) and from the 1st of July to the 2nd of August (see section Models
used to predict environmental conditions). This period was chosen because the GHG fluxes were measured
during this period and we did not want to extrapolate outside our main measurement campaign. Moreover, this
period represents the peak growing season of this region.
**2.2.3 Vegetation data**
We took images from $CH_4$ and $N_2O$ collars on the measurement day and used them to classify the dominant
vegetation to five distinct classes, following the Circumpolar Arctic Vegetation Map physiognomic
classification system (Walker et al. 2005) with minor modifications (Fig. 1). We used the following classes:
barren (< 10 % vegetation cover), meadow (graminoids and forbs), evergreen shrub, deciduous shrub, and
wetlands. We also utilized a larger dataset of 5820 vegetation descriptions from the study design to create the
vegetation type map.
We collected biomass samples from above-ground vascular plants using the clip-harvest method during late
peak season, between 17th of July and 10th of August. Samples were collected within the chamber collars, and
were oven-dried at 70°C for 48 h and weighed after drying. We refer to this variable as biomass (g dry-weight
m$^{-2}$).
**2.2.4 Soil sampling and analyses**
We measured the thickness of the organic and mineral soil layers using a metal probe reaching up to 80 cm
depth. We collected soil samples (ca. 1 dl) from the organic and mineral layers using metal soil core cylinders (4
- 6 cm Ø, 5 - 7 cm height) during August in 2016-2018. The organic samples were collected from the top soil,
and mineral samples directly below the organic layer which was on average 6.6 cm deep. Large roots were
excluded from the samples. The soil samples were freeze-dried and analysed in the Laboratory of Geosciences
and Geography and Laboratory of Forest Sciences (University of Helsinki). Bulk density (kg m$^{-3}$) was estimated
by dividing the dry weight by the sample volume. Soil organic layer pH was analyzed following ISO standard
10390. Total carbon and nitrogen content (C%, N%) analyses were done using Vario Elementar Micro cube and
Vario Elementar Max -analyzer (Elementar Analysensysteme GmbH, Germany). Prior to CN% analysis,
mineral samples were sieved through a 2 mm plastic sieve. Organic samples were homogenized by hammering
the material into smaller pieces.
Soils in this landscape are acidic and likely have a minimal amount of carbonates; consequently, we assumed
C% to equal organic C%. Soil organic carbon and nitrogen stocks were calculated for the entire soil horizon up
to 80 cm (in 95 % of plots soil depth was less than that). Some plots lacked CN% data (30 % of the plots), and
therefore, we used soil organic matter content estimated with the loss-on-ignition method according to SFS 3008
(1990). We utilized a similar stock calculation framework using the bulk density, layer depth, and C% and N%
data as in Kemppinen et al. (2021) except we used average bulk density and mineral C% estimates in each
vegetation type in case that information was missing in stock calculation.



Soil samples for dissolved organic carbon concentration analyses in dry soil were collected between the 5th and
14th of July 2018. After the collection, samples were stored at 4 °C and then dried at 60 °C for at least 5 days.
Extraction of dissolved organic carbon was done using pure water extractions with 0.5 to 3 grams of dried soil
added to 40 ml of water following the WEOC protocol from (Hensgens et al., 2021). Extracts were immediately
filtered (0.7µm) using glass fibre filters, diluted, acidified to remove inorganic carbon, and measured on a
Shimadzu TOC V-CPN analyzer set on the Nonpurgeable Organic Carbon mode. We refer to this variable as
dissolved organic carbon.
**2.2.5 Remotely sensed data**
Remotely sensed optical and light detection and ranging-based (LiDAR) data describing topographic,
vegetation, snow, and surficial deposit conditions was used for upscaling the in-situ measured environmental
variables (Fig. 2, Sect. S4 and Fig. S1).
**2.3 Statistical analyses**
We investigated the dependencies of GPP, ER, NEE, $CH_4$ flux, and $N_2O$ flux on environmental variables using
statistical analyses which included analysis of variance (ANOVA), and machine learning modeling and
prediction. We developed machine learning models, in which we 1) upscaled environmental data (annual soil
temperature, soil temperature, soil moisture, soil C/N, soil organic carbon stock, dissolved organic carbon,
biomass) using remotely sensed variables as predictors; 2) modeled GHG fluxes using the environmental data as
predictors, and 3) upscaled GHG fluxes using the upscaled environmental data maps at a 2-meter spatial
resolution across the landscape (Fig. 2). This two-step upscaling approach enabled us to focus on the
relationships between GHG fluxes with their physical and ecological, in-situ measured environmental controls
instead of the remotely sensed data that are proxies by nature. We ran all analysis in the R statistical
programming environment (R Core Team 2020; version 4.0.3).

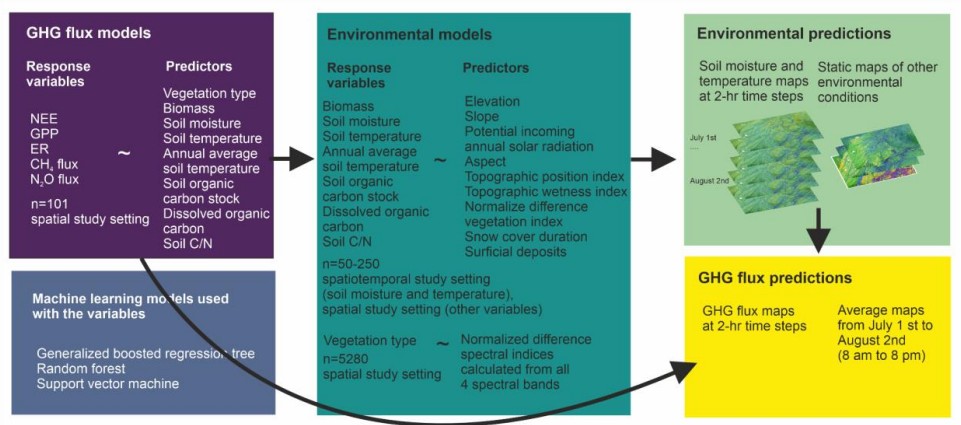






**Figure 2**: The upscaling framework used in this study. We first linked GHG fluxes to the in-situ environmental
drivers using machine learning models. Then we trained three machine learning models to upscale environmental
conditions across the landscape using remote sensing data. Then we used the GHG flux models and environmental
predictions to upscale GHG fluxes across the landscape throughout the entire growing season.
**2.3.1 Analysis of variance (ANOVA)**
We used one-way ANOVAs to test for vegetation type differences in environmental conditions, GHG fluxes,
and tested significance using multiple comparisons with a Tukey's honest significant difference test ($p < 0.05$).
$CH_4$ flux, soil moisture, soil organic carbon and nitrogen stock, and biomass were not normally distributed, thus
we used Kruskal-Wallis test instead of ANOVA at first.
**2.3.2 Machine learning models**
We modeled our response variables using three machine-learning methods (generalized boosted regression
models, GBM; random forest, RF, and support vector machine regression, SVM), all of which have been widely
used in flux upscaling studies (see e.g. Natali et al., 2019; Peltola et al., 2019; Tramontana et al., 2016). Based
on these models, we visualized the partial dependence plots characterizing the relationships between the
response and predictor variables while accounting for the average effect of the other predictors in the model
using the pdp package (Greenwell, 2017). Further, we calculated variable importance using the vip package
(Greenwell et al., 2020). Variable importance scores were estimated by randomly permuting the values of the
predictor in the training data and exploring how this influenced model performance based on the adjusted $R^2$
values, with the idea that random permutation would decrease model performance (Breiman, 2001). For more
details, see Sect. S5.
We used ten topography, snow, vegetation, and surficial deposits variables to construct landscape-wide
predictors matching the in-situ environmental conditions that we used to model the GHG flux values. These
variables were the following: elevation, topographic wetness index, topographic position index at 5 and 30 m
radii, aspect, slope, potential incoming solar radiation, normalized difference vegetation index, snow cover
duration, and surface deposits. Soil organic carbon stocks, dissolved organic carbon, soil C/N, biomass, and
annual soil temperature models were calibrated only once and a single prediction was made to the landscape.
Soil temperatures and moisture vary throughout the growing season, thus, we calibrated each model at each time
step and created 231 predictions over the study period (every 2 hours between 8 am and 8 pm from July 1st until
August 2nd). For each variable, an ensemble prediction was produced by calculating a median prediction over
the three predictions from the different modeling methods. Soil organic carbon stock was log+1 and biomass
were log-transformed prior to tuning the models, and after making the predictions, values were transformed
back to the original scale.
We examined the relationship between the five primary response variables (GPP, ER, NEE, $CH_4$ flux, $N_2O$ flux)
and environmental predictors that describe (i) soil resources and conditions (soil moisture, soil C/N, soil pH)
which are relevant to, for example, the growth of organisms (Nobrega and Grogan, 2008; Happonen et al.,
2022); (ii) soil C and N stocks and dissolved organic carbon which are one of the main sources for the GHG



emissions (Bradley-Cook and Virginia, 2018); (iii) soil temperatures which regulate enzymatic processes (St
Pierre et al., 2019; Mauritz et al., 2017); and (iv) biomass and vegetation type which describe resource-use
strategies, carbon inputs to soils and plant photosynthetic capacity, and integrate multiple environmental
properties into one variable (Magnani et al., 2022). We excluded soil pH and soil nitrogen stock from modeling
analyses due to high correlations (Pearsons's r>0.7) with soil moisture and soil organic carbon stock,
respectively. We did not use air temperature as a predictor as we already controlled for it in $CO_2$ fluxes in the
light-response model, and we assumed that soil microbes regulating $CH_4$ and $N_2O$ cycling are most importantly
driven by soil temperatures (Kuhn et al., 2021). The final predictors for our models were soil moisture, soil
temperature, annual soil temperature, soil organic carbon stock, dissolved organic carbon, soil C/N, biomass,
and vegetation type. After exploring the distribution of residuals of the preliminary GHG flux models, we
transformed $CH_4$ fluxes with cube-root transformation, and soil moisture with log transformation prior to tuning
the $CH_4$ flux model; in other models transformations were not necessary. The machine learning parameters
tuned for each model can be found from Sect. S5.
We used the machine learning models to predict GHG fluxes across the landscape for each 2-hour time step
from July 1st until August 2nd. Similar to the environmental predictions, an ensemble prediction was produced
by calculating a median prediction over the three predictions from the different modeling methods. As our focus
was on understanding the spatial patterns in the mean growing season fluxes, we averaged GHG flux predictions
over the study period. However, a visualization of the predicted mean daily patterns in soil moisture and
temperatures, and the consequent GHG fluxes is provided in the supplementary material (Fig. S2).
To compare the magnitude of all three important GHGs, namely $CO_2$, $CH_4$, and $N_2O$, we calculated the radiative
forcing strength of the three GHGs over a 100-year period from our measurements and ensemble predictions.
We used the Global Warming Potential (GWP; 27 for $CH_4$ and 273 for $N_2O$ (IPCC 2021)) and sustained GWP
(45 for $CH_4$ and 270 for $N_2O$ (Neubauer 2015), which are, to our knowledge, the best and most widely used
approaches that exist to compare flux magnitudes. We acknowledge that these approaches are designed to
quantify an effect of a change in emission to the radiative forcing, and are thus not fully suitable to be used to
quantify the climatic effect of natural continuous fluxes in our study (Mathijssen et al., 2022; Frolking et al.,

318    2006).


For all of our models, we used a leave-one-plot-out cross validation scheme in which each plot was iteratively
left out from the data set, and the remaining data were used to predict fluxes for the excluded plot to assess the
predictive performance of the models (Bodesheim et al., 2018). Estimates of bias were calculated as an average
of the absolute error (MAE) between prediction and actual observation. Coefficient of determination ($R^2$) was
used to determine the strength of the linear relationship between the observed and predicted fluxes.
The root mean squared error (RMSE) was used to estimate the model error. Uncertainty in GHG flux
predictions was derived by bootstrapping (fractional resampling with replacement based on vegetation type
classes). We subset the model training data into 30 different data sets, all of which had the same number of
observations as the original data itself. These 30 data sets were then used to produce 30 individual predictions
for a subset of the times with all three machine learning models and their ensemble for each response variable
(Sect. S5). The uncertainty estimates represent how different distributions of the input data as well as model
parameters influence the upscaled flux maps.






**3 Results**
**3.1 Environmental conditions and GHG fluxes across vegetation types**
We observed large variability in GHG fluxes and environmental conditions within and across vegetation types
(Fig. 3, Table S2). The variability in the different vegetation types differed depending on the flux and
environmental variable considered (e.g., meadow class had large variability in GPP and evergreen shrub class in
soil C/N). Frequently, wetlands differed clearly from the other vegetation types. While wetlands had high $CH_4$
emissions, all the other vegetation types with significantly lower soil moisture showed $CH_4$ uptake. Meadows
were a significantly larger net $CO_2$ sink than evergreen shrub sites, while other vegetation types had
intermediate NEE values. The $N_2O$ fluxes were low from all vegetation types, and varied between small sinks
and small sources.


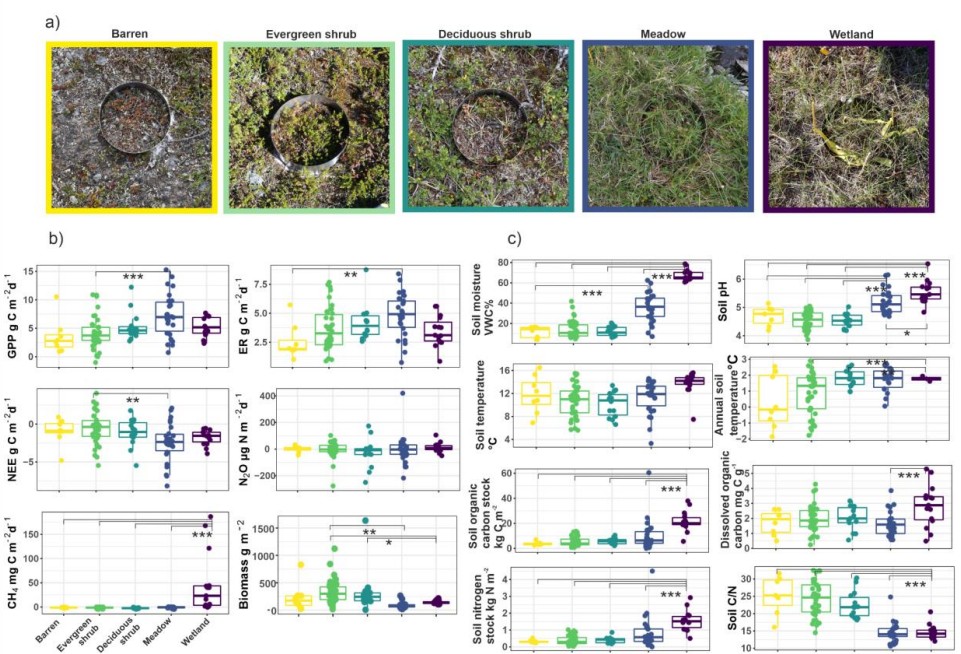


**Figure 3**: The vegetation types considered in this study (a), the distribution of GHG fluxes and biomass (b), and
environmental conditions (c) across the vegetation types. Lines represent Tukey's test results (* = p ≤ 0.05, ** =
p ≤ 0.01, *** = p ≤ 0.001). The box corresponds to the 25th and 75th percentiles, and the line within the box
represents the median. The lines denote the 1.5 IQR of the lower and higher quartile, where IQR is the inter-
quartile range, or distance between the first and third quartiles.





## 3.2 The performance of environmental and greenhouse gas flux models


The predictive performance of the ensemble environmental variable models was rather high but varied
depending on the variable ($R^2$: 0.43-0.71 except for soil temperature and soil dissolved organic carbon <0.26;
Fig. S3). The predictive performance of the GHG models was for most variables lower ($R^2$: 0.00-0.80), with
$N_2O$ flux models being close to random and $CH_4$ models performing the best (Fig. 4). The scatterplots of
observed and predicted GHG fluxes suggest that the highest flux estimates are often predicted most poorly, but
the mean fluxes in each vegetation type were predicted accurately.

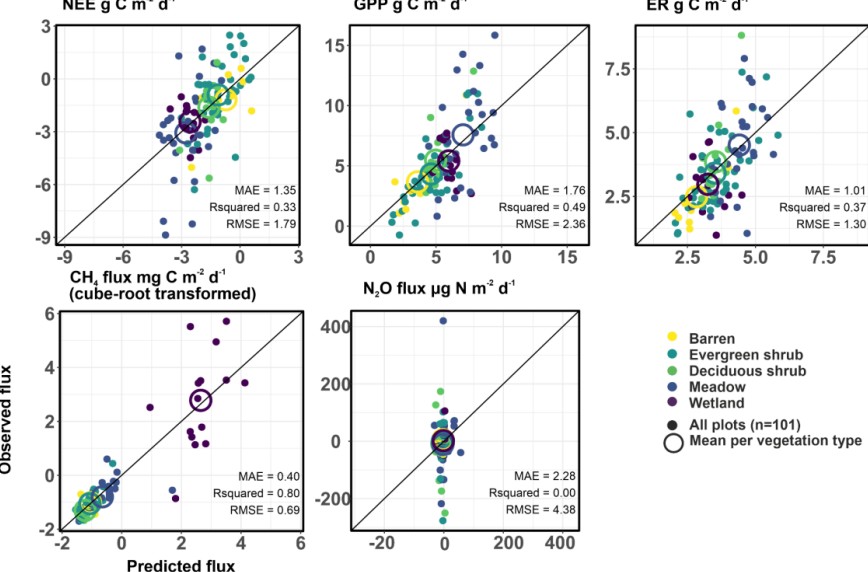



**Figure 4**: The correlation between observed and predicted values based on the ensemble model predictions (i.e.
median of the three machine learning model outputs). Model predictive performance is described with mean
absolute error (MAE), $R^2$ (Rsquared), and RMSE (root mean square error).

## 3.3 Drivers of greenhouse gas fluxes

The most important controlling variables and the response shapes differed depending on the GHG flux (Fig. 5,
Fig. 6 and Fig. S4), and sometimes also depending on the machine learning model type applied. $CO_2$ fluxes
were driven by annual average soil temperature, biomass, and vegetation type. In addition, soil organic carbon
stocks were an important predictor for ER. Soil moisture and vegetation type were the most important predictors
for $CH_4$ fluxes, and soil C/N and soil moisture for $N_2O$ fluxes. In general, warmer and wetter conditions
increased net emissions of $CH_4$ and $N_2O$ and net sink of $CO_2$. Some fluxes were further positively correlated
with soil organic carbon stocks (ER, $CH_4$ flux) and negatively with soil C/N (GPP, ER, $N_2O$).







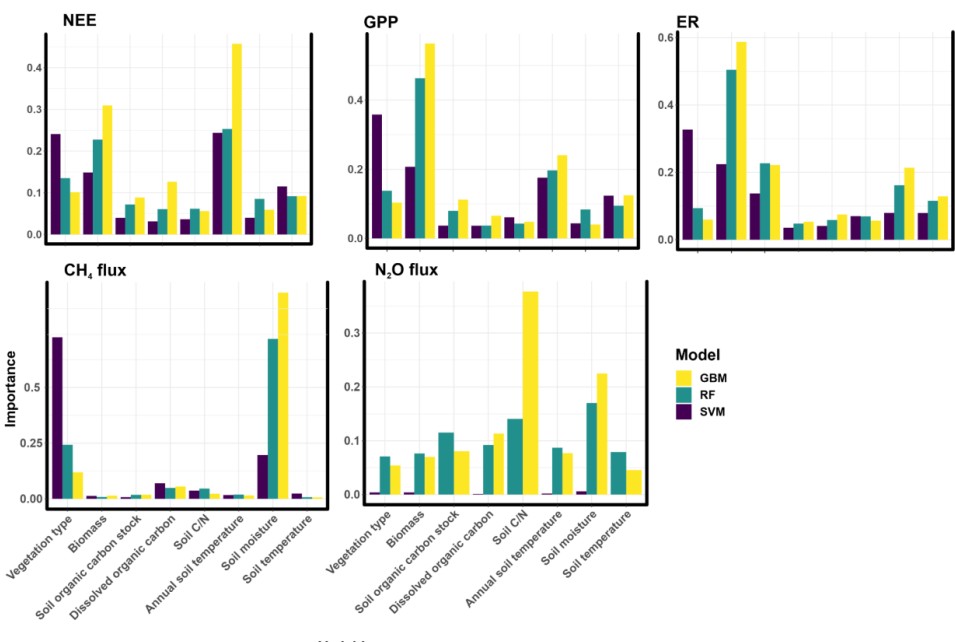


**Figure 5**: The variable importance of the environmental variables used to predict GHG fluxes. The models were
generalized boosted regression models (GBM), random forest (RF), and support vector machine regression
(SVM).










**Figure 6**: Partial dependence plots showing the relationships between GHG fluxes and environmental
conditions across the three models (generalized boosted regression models, GBM; random forest, RF; and
support vector machine regression, SVM). The y-axis of the plot (yhat) represents the marginal effect of the
predictor on the response and should not be directly compared with observed or predicted values, rather the
shape and direction of the response instead. Partial dependence plots for GPP and ER are found in Fig. S4.
**3.4 Spatial patterns and contributions in greenhouse gas flux predictions**
The model predictions show large spatial variability in GHG fluxes across the landscape (Fig. 7, Fig. S5). Net
$CO_2$ uptake as well as GPP and ER were highest in the warm and productive meadow locations of the valley
whereas $CH_4$ and $N_2O$ fluxes were highest in the eastern parts of the landscape that is dominated by wetlands.
The prediction suggests small but widespread net $CH_4$ uptake across the entire upland region. $CO_2$ was the most
important flux contributing to the net GHG sink (Fig. 8). Mean fluxes calculated based on the upscaled flux
maps differ from the in-situ based ones, particularly for wetland $CH_4$ emissions (Fig. 8, Fig. S6).



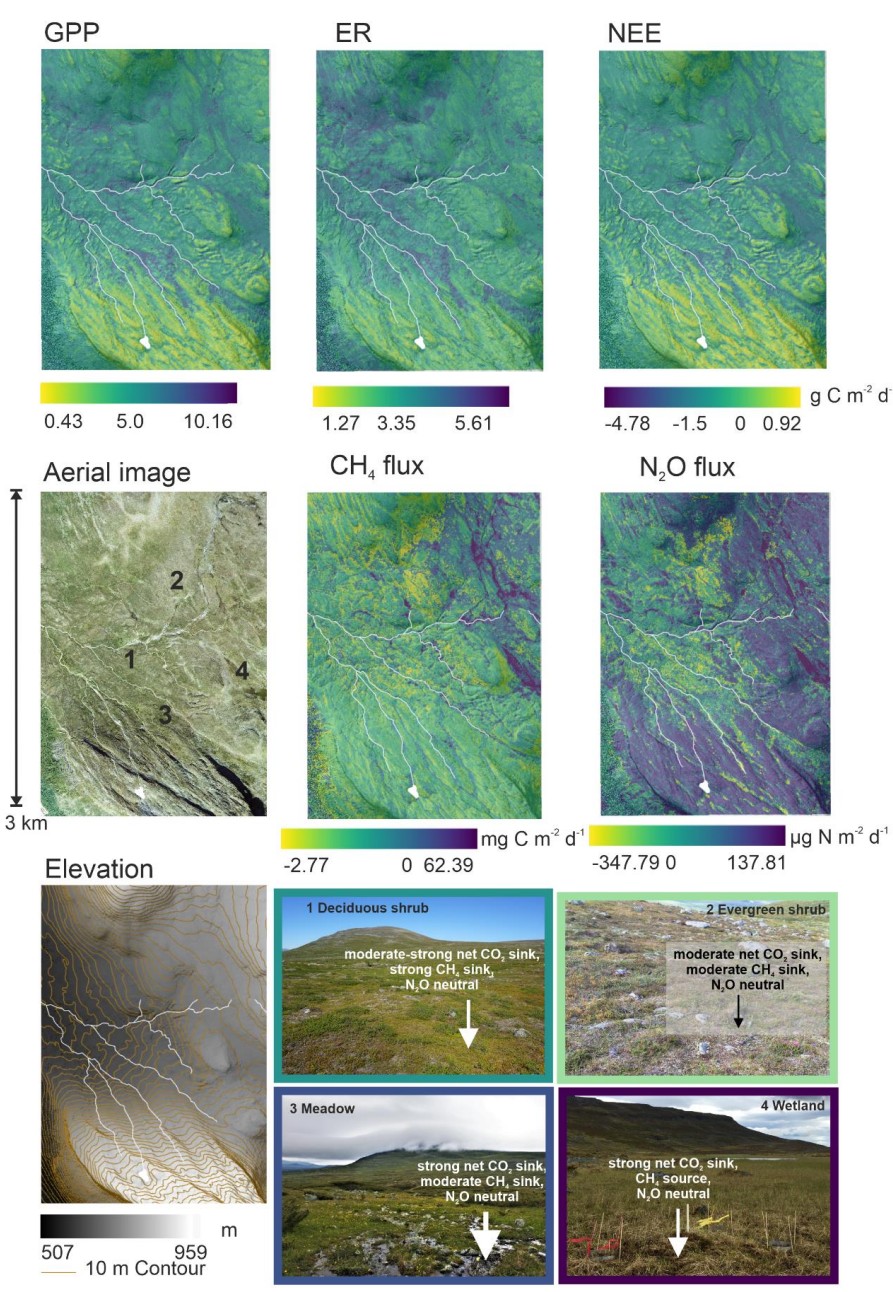





**Figure 7**: Ensemble predictions of growing season GHG fluxes, averaged over the 1st of July to the 2nd of
August (only daytime variability between 8 am and 8 pm considered) and photographs summarizing the main
sink-source patterns in the landscape. Note that the southwestern corner of the study design has mountain birch
forest for which we did not have any data; we did not have measurements from the northeastern corner either.




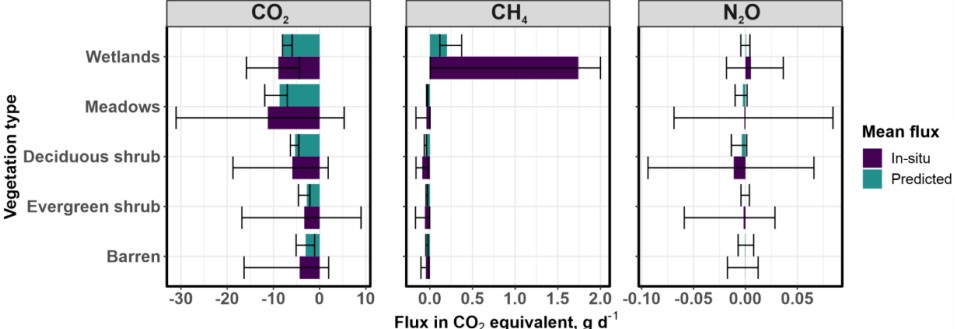


**Figure 8**: Growing season mean and percentile (0.025 and 0.975) GHG fluxes in $CO_2$ equivalents based on in-
situ data and upscaled flux predictions, averaged across the entire study period (only daytime variability
between 8 am and 8 pm considered) and across vegetation types. Note that the scale for the x axis is different for
each gas species, and that the uncertainties in in-situ versus predicted mean fluxes cannot be directly compared
with each other. The uncertainty in in-situ wetland $CH_4$ continues up to 6.7 but was cropped for visualization
purposes. The same graph using the sustained GWP approach can be found in the Supplement Fig. 6 and
demonstrates the potentially larger role of $CH_4$ fluxes over a 100-year horizon in this landscape.
**4 Discussion**
**4.1 $CO_2$ fluxes driven by both biotic and abiotic variables**
Our results show the importance of several environmental variables for $CO_2$ fluxes, demonstrating the strong
dependence of GPP and ER on a wide range of soil microclimatological, hydrological, soil biogeochemical, and
ecological processes (Sørensen et al., 2019; Dagg and Lafleur, 2011; Nobrega and Grogan, 2008; Cahoon et al.,
2016). Overall, the relationships with environmental conditions and GPP and ER were rather similar.
Aboveground plant biomass and vegetation type were important drivers for both which suggests a dominance of
autotrophic (plant) respiration over heterotrophic (microbial) respiration. Biomass was a more important
predictor than vegetation type for all the $CO_2$ fluxes, indicating that the quantity of plant material producing and
emitting carbon was potentially more important than the different types of plants associated with $CO_2$ cycling in
this study setting (Happonen et al., 2022). The high importance of plant-related variables (e.g., leaf area index)
as drivers of spatial variability in $CO_2$ fluxes has been previously found in other tundra landscapes (Marushchak
et al. 2013 and references therein).



Our models also show that annual soil temperatures have a different and stronger relationship with $CO_2$ fluxes
than instantaneous growing season soil temperatures, and these two soil temperature variables are indeed
negatively correlated in this study design. This is because annual soil temperatures are driven by winter soil
temperatures which increase with thicker snow cover that is found particularly in the valley and in
microtopographic depressions, which are colder in the summer. Moreover, annual soil temperatures integrate
many other environmental conditions from the entire year: they reflect growing season length and temperature
conditions, regulate C and N availability, and control vegetation and microbial community composition and
functioning over long time scales. These conditions have been shown to be important drivers of $CO_2$ fluxes
across a range of Arctic sites (Zona et al., 2022; Lund et al., 2010). Similar to these previous studies, we
observed that plots with warmer annual soil conditions have larger growing season GPP and ER fluxes and
stronger net uptake. Our results also show other logical relationships between environmental conditions and $CO_2$
fluxes. For example, GPP and ER increased with soil moisture (Nobrega and Grogan, 2008). However, at
around 50-60 % VWC this relationship plateaued and turned negative. This was likely due to the lack of oxygen
causing plants to suffer and microbes to produce $CH_4$ instead of $CO_2$ in methanogenesis (Bridgham et al., 2013).
Further, soil organic carbon stock was an important predictor for ER, but not so much for GPP. This was likely
related to the higher soil carbon contents boosting decomposition (Schlesinger and Andrews, 2000).

## 4.2 Small but consistent net $CH_4$ uptake mostly driven by soil moisture

Net $CH_4$ flux was strongly controlled by soil moisture due to its effect on regulating the anoxic and oxic soil
conditions, and therefore $CH_4$ production (methanogenesis) and $CH_4$ consumption ($CH_4$ oxidation, or
methanotrophy) (Kelsey et al., 2016; Christensen et al., 1996; Treat et al., 2018b). Our results demonstrate that
the rate of $CH_4$ emissions increases sharply in water-logged soil conditions, i.e. at soil moisture levels of >60
VWC%. In drier conditions (VWC < 60%), soils contain more oxygen, which prevents $CH_4$ production and
increases net $CH_4$ uptake. This result supports findings from recent studies that show that drier upland tundra
areas can be habitats for methane oxidizing bacteria which can use $CH_4$ from the atmosphere as their main
energy source, transforming these environments to net $CH_4$ sinks (Christiansen et al., 2015; Juncher Jørgensen
et al., 2015; Lau et al., 2015; Emmerton et al., 2014; Wagner et al., 2019; St Pierre et al., 2019). Given the large
area of the Arctic, even minor fluxes such as those observed here for $CH_4$ uptake can be of global importance.
This $CH_4$ uptake can strengthen the GHG sink of the Arctic and prevent $CH_4$ from entering the atmosphere.
Our results show that net $CH_4$ uptake increases not only in drier conditions but also in soils with low C/N, soil
dissolved organic carbon, and carbon stocks. This is likely due to microbes needing and getting C and energy
from the atmosphere due to limited soil C supply (Lau et al., 2015; Juutinen et al., 2022), and the capability of
methanotrophs to effectively compete against classical heterotrophs dependent on larger organic
macromolecules in these environments. The models did not clearly identify a particular vegetation type
controlling net $CH_4$ uptake, however some individual models demonstrated deciduous shrubs and meadows to
be more closely related to net $CH_4$ uptake (Larmola et al., 2010). Overall, our results indicate that net $CH_4$
uptake potential is present in any kind of upland tundra vegetation type (Fig. S7) as long as the abiotic
conditions for microbes responsible for atmospheric $CH_4$ consumption are favourable.





CH$_4$ fluxes had a rather uniform distribution across the mineral upland regions (i.e., small but consistent net
uptake). High CH$_4$ emissions were located in wetland regions dominated by high soil organic carbon stocks and
moisture levels. Our observations demonstrated similar, or even higher net CH$_4$ uptake than previous studies.
For example, dry tundra was CH$_4$ neutral in a recent Arctic-Boreal CH$_4$ flux synthesis (mean=3.83, median= -
0.01 mg CH$_4$ m$^{-2}$ d$^{-1}$; primarily based on growing season daytime fluxes; Kuhn et al., 2021) whereas our study
showed higher uptake rates for the non-wetland plots (mean=-2.05 , median=-1.81 mg CH$_4$ m$^{-2}$ d$^{-1}$). However,
studies focusing on individual sites have recorded similar CH$_4$ flux magnitudes as observed here (Emmerton et
al., 2014; Lau et al., 2015), but to the best of our knowledge, such extensive spatial patterns in CH$_4$ flux uptake
using fine spatial resolution models as presented here have not been published so far.
**4.3 N$_2$O fluxes remain neglectable and unpredictable**
We observed moderate, and to a large extent unpredictable variability in N$_2$O fluxes in this landscape. The
differences in average fluxes between the vegetation types were small. Based on our observations, most
vegetation types were on average N$_2$O sinks or neutral but deciduous and evergreen shrubs and meadows had
some variability from moderate N$_2$O sinks (up to -300 µg N m$^{-2}$ d$^{-1}$) to moderate N$_2$O sources (up to 400 µg N
m$^{-2}$ d$^{-1}$). Overall, our average N$_2$O fluxes were close to zero and thus low in the light of the recent review (Voigt
et al., 2020), which demonstrated that vegetated soils in permafrost regions are often small but evident sources
of N$_2$O during the growing season (~30 µg N m$^{-2}$ d$^{-1}$), and that barren or sparsely vegetated soils serve as
substantial sources of N$_2$O (~455 µg N m$^{-2}$ d$^{-1}$). The relatively small N$_2$O fluxes observed here can be explained
by the nitrogen-limited nature of the studied soils and the strong competition between plants and microbes for
nutrients: with low nitrogen stocks, nitrogen release by mineralization remains low (Voigt et al., 2020). Another
possible reason for the difference between our results and the synthesized N$_2$O flux estimates in Voigt et al.
2020 is that most of the data in the synthesis came from ecosystems that are not as much nitrogen-limited as our
site (e.g., peatlands, grasslands).

We were unable to explain the patterns in N$_2$O fluxes with the predictors used here. This was likely related to
the relatively low variability in N$_2$O fluxes in most of the plots in general, and the complexity of the soil
microbial N cycle, where N$_2$O is produced (nitrification, denitrification, DNRA) and consumed (denitrification)
by multiple co-occurring processes, differently regulated by environmental variables (Butterbach-Bahl et al.,
2013). Nevertheless, the most important driver of N$_2$O flux was soil C/N, and the models suggested that lower
C/N ratios were linked to higher net N$_2$O emissions. This is potentially due to the excess soil N allowing for
more rapid N mineralization, nitrification and denitrification which accelerate N$_2$O emissions (Klemedtsson et
al., 2005; Liimatainen et al., 2018). Further, N$_2$O emissions were highest in the wetlands, similar to Ma et al.,
2007 who explained this by high ammonia or nitrate levels boosting N$_2$O production. The uppermost soil layers
were also likely not fully saturated by water at the time of the wetland measurements, which can induce higher
N$_2$O emissions in oxic conditions (Voigt et al., 2020; Takakai et al., 2008). In contrast to C fluxes, vegetation
type did not play an important role for N$_2$O fluxes. This might be related to our study having no measurements
in the previously observed, clear N$_2$O flux hot spots located in barren permafrost peatlands, such as peat
plateaus or palsas, with thick organic layers and high inorganic N content (Repo et al., 2009; Voigt et al.,
2017a).



### 4.4 The sub-Arctic tundra landscape is a strong growing season GHG sink


Our results demonstrate a high level of spatial heterogeneity in the growing season GHG fluxes across the
landscape, with all three gases acting as both net sinks and sources in some parts of it. Areas acting as GHG
sinks covered most of the landscape ($CO_2$: 92 %, $CH_4$: 95 %, $N_2O$: 64 %; 61 % of the area was a sink for all the
three GHGs). We observed clear differences in flux magnitudes driven by key environmental conditions. Moist,
and carbon and nitrogen-rich meadows and deciduous shrub heaths were strong GHG sinks. Wet sedge-
dominated fens were GHG sinks with $CH_4$ emissions being compensated by net $CO_2$ uptake. Barren lands and
evergreen shrubs were more resource-limited and closer to GHG neutral. These results are interesting in the
light of the shrubification patterns observed across the entire Arctic (Myers-Smith et al., 2011; Parker et al.,
2015; Vowles and Björk, 2018), and indicate that evergreen or deciduous shrub expansion may increase or
decrease the growing season GHG sink. If shrubs expand to meadows, the GHG sink may decrease, whereas if
they invade barren areas, the GHG sink may increase. However, our results did not quantify this change over
time, or cover the entire year to confirm the net annual effect.

Our results indicate that this heterogeneous Arctic landscape was a cumulative net GHG sink during the
measurement period during daytime (8 am to 8 pm) in July 2018. The July budget for $CO_2$ was -4.6 g C m$^{-2}$
month$^{-1}$, for $CH_4$ -3.7 mg C m$^{-2}$ month$^{-1}$ and for $N_2O$ -12.9 µg N m$^{-2}$ month$^{-1}$. The $CO_2$ sink is relatively small,
likely due to the high cover of patchy and sparsely vegetated areas that were often $CO_2$ sources. This small sink
value is likely an overestimation as we did not have measurements from the night time and did thus not upscale
fluxes in night-time conditions when ecosystems are net $CO_2$ sources due to the lack of light required for
photosynthesis. It also overestimates the importance of $CO_2$ as a radiative forcing agent, since ecosystem $CO_2$
production during autumn and winter contributes substantially to the annual C balance (Celis et al., 2017;
Commane et al., 2017), thereby reducing the $CO_2$ sink strength on an annual basis. Further, $CH_4$ uptake might
continue even in rather cold conditions as long as soils remain dry and unfrozen (Emmerton et al., 2014).
Nevertheless, our results demonstrate that net $CO_2$ uptake plays the most important role for the net growing
season GHG budget, but a small proportion of the GHG sink strength during growing season originates from net
$CH_4$ uptake. The role of $N_2O$ fluxes for the net GHG budget across the entire landscape is negligible for the
growing season.

### 4.5 Methodological considerations in GHG flux modeling


Our study creates new understanding about high-resolution upscaling of GHG fluxes by incorporating more
chamber measurements, predictors, models, and environmental gradients compared to earlier efforts (see e.g.,
Fox et al., 2008; Dinsmore et al., 2017; Räsänen et al., 2021; Juutinen et al., 2022; Vainio et al., 2021). For
example, we included chamber measurements from 101 plots whereas earlier local-scale upscaling studies have
usually had circa 30 plots. Further, we included eight different environmental predictors while other studies
have often used only one or two, focusing on predictors describing vegetation type or soil moisture. Finally, we
studied a tundra landscape that consists of almost all the main vegetation types of the entire Arctic, whereas
earlier studies have investigated a narrower range of vegetation conditions, with a focus on wet ecosystems.



Our study showed that using means of in-situ GHG fluxes in each vegetation class to derive a landscape-level
GHG budget might produce significantly different results compared to the upscaled budget. This was apparent
particularly for $CH_4$ fluxes, where the in-situ based average wetland $CH_4$ emission was more than seven times
larger $CH_4$ compared to the upscaled one. This mismatch is likely explained by the heterogeneity of
environmental conditions and $CH_4$ fluxes within the wetland class that the chamber measurements alone could
not cover (Fig. S8). A multivariate machine learning modeling approach with variables describing not only
vegetation type but also soil moisture and other conditions were likely able to characterize the resulting $CH_4$
flux variability in a more representative way. For example, our soil moisture maps showed high variation in soil
moisture between ca. 50 and 70 VWC % within the wetland areas, and high $CH_4$ emissions were observed only
in areas with 60 VWC %. Overall, this result suggests that simple land cover-based upscaling efforts might lead
to biased budget estimates, especially when spatial variability within land cover types is high, emphasizing the
need for multivariate models in flux upscaling.
The performance of our models varied from good (GPP, $CH_4$ flux), moderate (ER and NEE) to low ($N_2O$). $CH_4$
fluxes - both sources and sinks - were most accurately modeled, providing important support for future studies
predicting not only the large $CH_4$ emissions but also the previously unquantified $CH_4$ uptake in Arctic
landscapes. The lower predictive performance of the models for other GHG fluxes might be explained by the
dynamic nature of fluxes not being represented in our spatial study design, and our models lacking important
predictors. The performance of the models could potentially be improved by describing plant functional
composition using plant traits (Happonen et al., 2022), and including more detailed information about soil
nutrients (e.g., soil nitrate or ammonium concentrations as soil C/N captures only very roughly how much N is
available) or microbial communities (e.g., communities or genes associated with nitrification or methanogenesis
or methanotrophy; Pessi et al., 2022).
We chose to use in-situ environmental data as predictors of GHG fluxes in our upscaling framework instead of
linking remotely sensed variables with GHG fluxes directly. This was done to increase understanding about the
mechanistic and ecological relationships but required us to first produce spatially continuous maps of
environmental conditions, which might have added an additional layer of uncertainty into our framework.
However, the most important environmental variables (i.e., soil moisture, temperature, biomass) had a high
predictive performance. Nevertheless, future studies could explore the performance and information derived by
upscaling GHG fluxes using high-resolution satellite or drone-derived remotely sensed indices directly (Siewert
and Olofsson, 2020; Vainio et al., 2021; Berner et al., 2018).
Overall, the performance of our machine learning models predicting spatial variability in GHG fluxes was
weaker than in other studies focusing on temporal variability (e.g., López-Blanco et al., 2017; Celis et al., 2017),
even though we had a comprehensive set of environmental measurements. Our results thus highlight the need
for more focus on the spatial patterns in GHG fluxes. While the temporal variability is widely acknowledged as
a source of uncertainty in GHG budget estimates (Baldocchi et al., 2018), the spatial variability may be just as
important but remains insufficiently studied (Treat et al., 2018c). Study designs focusing on spatial variation in
GHG fluxes using a combination of intensive measurement campaigns, remotely sensed datasets, and modeling





approaches are informative although they do not produce direct information on the trends and drivers of GHG
flux change following climate change. They provide new knowledge about the heterogeneity in GHG fluxes and
their environmental drivers which is highly important for understanding flux magnitudes from local to global
scales. Further, they can be used as a space-for-time substitution to understand ecosystem functions in locations
that are assumed to be at different stages of development. Moreover, this knowledge is valuable for designing
representative field studies in the future.

**5 Conclusions**
This study showed that predicting fluxes in heterogeneous tundra landscapes at high spatial resolutions is
possible for $CH_4$, GPP, and to some extent also NEE and ER fluxes but remains a challenge for $N_2O$ fluxes. This
is a promising result for future high spatial resolution modeling studies that aim to understand the fine-scale
biogeochemistry of the rapidly changing Arctic environments. Our study further demonstrates high spatial
variability of GHG fluxes which is driven by a multitude of vegetation, soil microclimatological, hydrological,
and biogeochemical conditions. The upscaling shows the importance of net $CO_2$ uptake for the peak growing
season net GHG budget, and suggests that annual soil temperature and vegetation parameters are the most
important drivers. Most importantly, it reveals small but widespread $CH_4$ uptake across the entire upland tundra
in our domain that switches the studied landscape, consisting of wetlands with high $CH_4$ emissions, to a small
net $CH_4$ sink. This provides more evidence to the relatively unquantified but important $CH_4$ sink in the Arctic
GHG budget.
**Code/Data availability**
The field data, analysis codes and most of the results are available in a repository (Virkkala et al. 2023).
Upscaling results for each individual timestep were not included in the repository due to their large size, but
they can be acquired from the author upon request.
**Author contribution**
AMV and ML conceptualized the research with input from PN, JK, MEM, and CV. AMV, PN, JK, MEM, JK,
CV, GH, VT, JH and ML contributed to data collection. AMV analyzed the data and wrote the manuscript draft.
All the coauthors reviewed and edited the manuscript.
**Competing interests**
The authors declare no competing interests.



**Acknowledgements**


The authors would like to acknowledge the support by the research assistants during the data collection as well
as Kilpisjärvi Biological Station. AMV was supported by The Finnish Cultural Foundation, Alfred Kordelin
Foundation, Väisälä fund, and Jenny and Antti Wihuri Foundation, and the Gordon and Betty Moore foundation
(grant #8414). AMV and ML acknowledge the Academy of Finland funding (grant #286950). AMV and GH
acknowledge the Svenska Sällskapet för Antropologi och Geografi for funding. C.V. was supported by the
Academy of Finland project MUFFIN (no. 332196). ML acknowledges Academy of Finland funding (grant
#342890). C.B acknowledges funding from Academy of Finland general research grant (project N-PERM,
decision Nr. 341348). CB, CV, and MEM acknowledge Academy of Finland/Russian Foundation for Basic
Research project NOCA (decision no. 314630). PN was funded by the Academy of Finland (project number
347558). JK was funded by the Academy of Finland (project number 349606). JH was funded by the Academy
of Finland (grant #308128). We acknowledge funding for fieldwork and equipment by the Nordenskiöld
samfundet, Tiina and Antti Herlin foundation, and Maa- ja vesitekniikan tuki ry.

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
