# Peer review of "High-resolution spatial patterns and drivers of terrestrial"

_Biogeosciences, 2023_

## Referee Comment (RC2)

**Review of "High-resolution spatial patterns and drivers of terrestrial ecosystem carbon dioxide, methane, and nitrous oxide fluxes in the tundra"**

**General Comments**

The authors use flux observations from 101 chamber plots and a dataset of environmental drivers to estimate daytime CO2, CH4 and N2O fluxes across a study site in northern Finland over July 2018. This dataset is unique and the topic is certainly within the scope of the Biogeosciences. I think the manuscript could make a valuable contribution to the literature on high latitude greenhouse gas fluxes. However, I feel the article requires major revisions to address some key pitfalls before it can be published.

While there are a large number of sample plots, the lack of temporal replicates concerns me. I understand the difficulty and expense associated with Arctic research, so I do not feel this issue alone should disqualify the manuscript, but this limitation should be discussed more in depth. The authors briefly address this but I feel it necessitates further explanation within the text itself.

- Sampling spanned only two days, between 10 am to 5 pm. Given this - I have a hard time believing "the spatial variation in our plots covered most of the temperature variation during the growing season" without a more thorough discussion.

  - Sec S1 gives mean air & soil temperature for the chambers during observation and over the study period. I would like to see these broken down in more detail, **with soil moisture too**. Perhaps as a boxplot in the supplement?
  - I assume these samples were collected under clear weather conditions. Fluxes during and after any rainfall events would be quite different. Was there much rain in July 2018? Perhaps rainfall days should be excluded from upscaling? Is it possible an upland site that is otherwise a sink could shift to a CH4 source during/after rainfall?

Perhaps I missed it but - How many plots there were for each vegetation type?

- Were samples sizes even between types? Weighted by spatial coverage?

I am concerned by the use of regression forest methods a dataset of this size. With 10 inputs, but only 101 flux samples (no temporal), it seems to me these models are severely over parameterized. I doubt that there are sufficient training samples for the models to adequately parse out the functional relationships in 10D feature space. It might be beneficial to consider pruning your model - you could use the feature importance to inform your choice of which variables to keep/remove. This would likely result in a more robust model that is less likely to produce spurious results.

- Random forest models are poorly suited for projection, often performing worse than simple linear regression (Hengl et al. 2018).
  - Hengl, T., Nussbaum, M., Wright, M., Heuvelink, G., & Graeler, B. (2018). Random forest as a generic framework for predictive modeling of spatial and spatio-temporal variables. PeerJ, 6, e5518. https://doi.org/10.7717/peerj.5518
- Looking a the partial dependence plots, the support vector machine appears to produce more reasonable results and it would be nice to see some discussion of why.
- I would like to see the authors incorporate a simpler method like ordinary least squares regression to their ensemble. I would also like to see a breakdown of the upscaled estimates for each model, in addition to the estimates for the ensemble median.

**Specific Comments**

**Introduction**

**Line 60:** "and they have different spatiotemporal dynamics with each other and compared to CO2 fluxes" - reads weird, consider rewording? "All three gasses have distinct spatiotemporal dynamics."

**Line 66:** Close parenthesis

**Materials and Methods**

**Line 93:** above *a* mountain birch

**Line 105 - 106:** "101 GHG flux measurement plots and 50 to 5280 plots with other environmental data" - this is confusing? 50 to 5280 plots? please explain better.

**Line 121:** Consider rephrasing - from Table S1 it looks like most factors had near complete coverage, so maybe say something like: "Environmental conditions explaining these GHG

fluxes were measured at each plot. Most environmental variables had near complete spatial coverage; missing data were filled using the environmental predictions"

**Line 168:** "Five gas samples were taken within a 50-min enclosure time" - this seems like a very long sampling interval? Are you concerned about heating within the chamber during the 50 min closure time disconnecting processes within the chamber from ambient conditions? Or about underestimating fluxes from high emitting wetland plots due to a reduction in the gas concentration gradient between the soil and the chamber head space? What was the rational for using this long sampling interval?

**Line 207-208:** "We also utilized a larger dataset of 5820 vegetation descriptions from the study design to create the vegetation type map. Please elaborate on how this data and how the map was created. Was this product created by a previous study? If so we need the citation. If not, you need give a more detailed description of the methods and data used to create the map.

**Line 239:** I don't think one sentence necessitates its own sub-section. Perhaps expand on this a bit or merge it with section 2.2.3

**Lines 266-268:** I feel this is insufficient justification for the choice of models. I would like to see a bit more on why these methods were chosen and the pros/cons of each.

**Line 302:** Seems like a reasonable thing to do - I assume the idea is to minimize the effect of outliers on the model? Perhaps explicitly say that in the text?

**Line 320:** Leave one out is concerning?

**Results**

**Lines 356-358:** "The scatter plots of observed and predicted GHG fluxes suggest that the highest flux estimates are often predicted most poorly, but the mean fluxes in each vegetation type were predicted accurately."

- This seems like an obvious point - empirical models will tend toward the mean of the training domain regardless of how well fit the distribution of the individual training points.

**Line 371:** net *uptake* of CO2

**Discussion**

**416-417:** "Aboveground plant biomass and vegetation type were important drivers for both which suggests a dominance of autotrophic (plant) respiration over heterotrophic (microbial) respiration." - this statement seems like a bit of a stretch? I would assume more above ground biomass also means more litter input for decomposition, and also would likely be correlated with below ground biomass » leading to greater microbial decomposition of root exudates? How strongly correlated were the input parameters?

**Lines 459-461:** Out of curiosity, for what portion of the year do you expect these favorable conditions to last? I'd imagine some of the sinks, especially the valley bottom meadow would be sources during snow-melt period, and possibly again during the freeze up period in fall?

**Line 503:** The gasses themselves do not act as sinks/sources, consider rephrasing.

**Line 510-511:** "evergreen or deciduous shrub expansion may increase or decrease the growing season GHG sink" - consider switching to "deciduous or evergreen shrub expansion may …" to better get your point across.

**Line 536:** perhaps a bit of a stretch to say "all the main vegetation types"?

**Line 543:** would the temporal variability of soil moisture and temperature contribute to the difference too? What was the variability like over the period - relative to the sample days? It would be nice to se a time series of soil temperature and moisture (averaged by vegetation type) over the July 2018 study period - with the sample dates/times highlighted for reference.

**Line 556:** or models having too many parameters …

**Line 575-577:** An important point, well put!

**Figures**

**Figure 1:** Plots a. and c. color schemes are misleading - using the same colors to show very different phenomena. I suggest changing the color for the vegetation maps to one better suited for discrete qualitative data. For plot b. the chambers should be color-coded by vegetation type so the reader can better see the spatial distribution of each type. Additionally, it would be helpful if the figure caption (or somewhere else in the text) said how many chambers there were for each vegetation type. For the numeric data in c. soil temperature and annual soil temperature are the colormaps are inverted relative to the other plots, which also makes things a bit unclear at first glance.

**Figure 3:** Could you make the boxplot larger so they're easier to read and consider excluding the points that are within the boxes - makes for a confusing/overly complex boxplot. Additionally:

- You could set the y-axis limits for GPP and ER to the same values for a more direct comparison.
- You have a "-0" on your boxplot for biomass
- Maybe just keep the most important plots and move some to the appendix to save space?

**Figure 5:** Are these importance values normalized to a 0-1 scale?

- Should the bars sum to 1 for each model?
- Why is the importance for N2O so low for SVM across all features
- Please use a common y-axis range across the subplots for easier comparison
- Any estimate of uncertainty in these feature importance estimates? e.g., for a RF model you can get the importance of each sub-model and use it to calculate a 95% CI over around the RF feature importance values.

**Figure 6:** It seems to me this plot highlights how regression trees (RF and GBM) models are poorly suited for this type of analysis, particularly given the small sample size. The noisy response curves they generate are because regression trees are treating each terminal node in a tree as a discrete data point to match rather than a continuous response function to fit. The idea behind a random forest, is that averaging many of these over-fit trees will emulate the desired response function (*only within the bounds of the training data*). However, the there do not appear to be sufficient samples for the RF model to be able to do that.

- Are these partial dependence yhat values in the same unit/scale for each predictor? If so it would be useful to have each y-axis on the same scale to see the relative magnitude of the partial dependence by variable.

**Figure 7:** Using "strong" and "moderate" to describe the CH4 sink strength of upland areas seems odd. By area, yes they may be large sinks overall, but on a per unit area basis they are not given their relatively small magnitude compared to wetland CH4 emissions shown in Fig 8. Perhaps try rephrasing the labels in the images? Alternatively, show your landscape map here to emphasize that you're talking on a "per landscape fraction" basis.

**Figure 8:** I would like to see a different color for the error bars to help them stand out from the plots more.

**Supplement:**

**Figure S4:** Needs a legend for the plots.

---

## Author Comment (AC1)

**Referee comments 1 RC1**

This study provides excellent insight into spatial heterogeneity of GHG fluxes in a fairly unprecedented sampling extent of chamber-based fluxes. The authors compensate well for the lack of temporal coverage in fluxes by leveraging the spatial information available through a hierarchical Bayesian modeling approach to NEE. The remote sensing analysis is thorough. The comparison of scaling models provides useful insight into important multivariate influences on GHG fluxes.

*Response: Thank you for this feedback!*

I have two areas of significant improvement to address. The first is a lack of information provided about the Bayesian hierarchical modeling. The authors reference Williams et al. 2006 for their model structure. This should be explicitly provided, along with the parameters that are being estimated, at least in the supplement. The authors mention using vague prior information for these parameters. The prior distributions and initial values used for the MCMC chains should be provided (at least in the supplement). There is no mention of posterior predictive checks or tests for convergence. These are necessary to ensure the model is appropriate for the data and that parameters are estimated correctly (without need for a longer burn-in for example). There should be some presentation of the posterior distributions for parameter estimates. There could be some additional discussion as well related to how much the random effect of plot contributed predictions, or how variable the random effect was within vegetation types, etc.

*Response: Thank you for pointing this out. We will clarify that ER, maximum GPP, half-saturation constant, and an exponential air temperature response of ER were the parameters in the main text.*

*We will add the following details to the Supplement:*

*"We used weakly informative priors for our parameters, informed by those reported in Williams et al. 2006 and Happonen et al 2022. The means and standard deviations for the priors were 1 and 2 for the logarithm of the ER (posterior: 0.65 and 0.51), 0 and 1 for the temperature effect on logarithm of the ER (posterior: 0.02), 10 and 10 for maximum GPP (posterior: 8.54 and 5.51), and 6.2 and 0.3 for the logarithm of the half-saturation parameter (posterior: 5.92 and 0.48). We used logarithms for some parameters to normalize their error distribution.*

*The Bayesian $R^2$ of the model was 0.96, four out of five family- and population-specific mean parameters had an Rhat of 1.00, and posterior predictive draws matched well with the observations (Fig. X.), indicating model convergence and good predictive performance. For more details on the model structure, see Happonen et al. 2022 (section 2.4.1) and the code light_response_model.R in Virkkala et al. 2023."*

[Figure]

*Fig. X. The distribution of NEE observations (y) and draws from the posterior predictive distribution (yrep) in micromoles $CO_2$ $m^{-2}$ $s^{-1}$.*

The second area for improvement is related to using back-transformations of log-transformed predictions. From my understanding, soil C, biomass, and soil moisture were log-transformed during their upscaling. Then they were back-transformed and subsequently used as drivers to predict GHG fluxes. Back transforming a prediction (from a non-affine transformation) will introduce bias that needs to be corrected. For a useful explanation of the problem, see this blogpost: https://florianwilhelm.info/2020/05/honey_i_shrunk_the_target_variable/. There are multiple methods available for correcting back-transformation bias, some of which are analytical such as in the case of simple linear regression. See this paper for a comparison of several bias correction methods for GBM models: https://kdd-milets.github.io/milets2022/papers/MILETS_2022_paper_0925.pdf. Since all three of the back-transformed variables rank as fairly high predictors, and are especially important at high soil C, high soil moisture, etc where the back-transformation bias is larger, it is critical to correct this bias. The CH4 flux scaling similarly needs a back transformation bias correction, since a cube-root transformation is also non-affine, and these predicted fluxes are subsequently back transformed for comparing to in situ fluxes and calculating carbon budgets.

*Response: We agree that back-transformations are problematic because they impact both the error distribution and the shape of the regression. While this can certainly be an issue for the environmental datasets (soil C, biomass, and soil moisture), we think that the largest uncertainties in our study would be with the back-transformation of the CH4 flux variable as it was one of our main GHG flux response variables. We will analyze how these transformations impact our main conclusions as well as explore potential ways to correct for the biases in the following ways:*

*1) For the transformed variables, we will compare the range of values in the observations and predictions to understand if the summary statistics (e.g., maximum and minimum values) in each vegetation type are different to see if there are potential issues.*

*2) We will test how the CH4 flux predictions change if no CH4 flux transformations are being made. Technically, machine learning models should be more flexible with the distributions and assumptions than linear regressions and therefore might work fine without transformations. We will compare the distribution of residuals, model predictive performances, and the resulting predictions.*

*3) We will test correcting for the bias with a non-linear model.*

*Based on these tests, we will either 1) keep the transformed variables as they are with evidence highlighting that backtransformations did not impact our final conclusions, 2) apply a bias-correction as suggested by the referee, or 3) remove all transformations after a careful re-examination of the model performance of the models without transformations.*

Minor comments:
There are numerous regressions demonstrating model performance (Fig 4, FigS3), with the r-squared reported. The slopes intercepts, and p-values should also be reported, as this would help assess performance and bias in the model predictions.

*Response: We will add these statistics to the manuscript.*

Font sizes for Fig 3 are too small.

*Response: We will reorganize the figure and remove some less important boxplots to make the fonts bigger.*

It is unclear what the 'Agency 2017' reference is in Fig 1. It is also unclear what the colored vegetation boxes correspond to in panel (c) of Fig 1.

*Response: We will correct and clarify these.*

---

## Author Comment (AC2)

**Referee comments 2 RC2**

**General Comments**

The authors use flux observations from 101 chamber plots and a dataset of environmental drivers to estimate daytime CO2, CH4 and N2O fluxes across a study site in northern Finland over July 2018. This dataset is unique and the topic is certainly within the scope of the Biogeosciences. I think the manuscript could make a valuable contribution to the literature on high latitude greenhouse gas fluxes. However, I feel the article requires major revisions to address some key pitfalls before it can be published.

While there are a large number of sample plots, the lack of temporal replicates concerns me. I understand the difficulty and expense associated with Arctic research, so I do not feel this issue alone should disqualify the manuscript, but this limitation should be discussed more in depth. The authors briefly address this but I feel it necessitates further explanation within the text itself.

*Response: Thank you for this feedback. We agree that in an ideal world, we would have study designs covering both the fine-scale spatial heterogeneity as well as the temporal variability, complemented by more controlled experiments to verify the drivers of change. In this study, however, we were primarily focusing on the spatial variation, as we had identified the lack of extensive spatial study settings from the literature. Doing the GHG flux chamber measurements alone took one month for 3 researchers, and unfortunately we did not have the resources to continue this throughout several months. We will add sentences to the main text emphasizing that spatial study designs can be used to infer correlations between variables, but correlation does not imply causation. We will also discuss if and how the relationships that we observe across spatial gradients compare with those observed in time-series studies where causal relationships can be more easily observed.*

- Sampling spanned only two days, between 10 am to 5 pm. Given this - I have a hard time believing "the spatial variation in our plots covered most of the temperature variation during the growing season" without a more thorough discussion.

  – Sec S1 gives mean air & soil temperature for the chambers during observation and over the study period. I would like to see these broken down in more detail, **with soil moisture too**. Perhaps as a boxplot in the supplement?
  – I assume these samples were collected under clear weather conditions. Fluxes during and after any rainfall events would be quite different. Was there much rain in July 2018? Perhaps rainfall days should be excluded from upscaling? Is it possible an upland site that is otherwise a sink could shift to a CH4 source during/after rainfall?

*Response: Thank you for pointing out the importance of the temporal representativeness of the data. The sampling spanned a month, from July 1st to August 2nd. The snow melts in May-June and plants reach their maximum biomass in July-early August, after which the autumn and senescence slowly start. We assume that clarifying this misunderstanding likely solves part of the issues raised by the referee in this comment.*

*We will add the following description to the Supplement: "Mean soil moisture was 27 %
during the CH4 and N2O flux measurements and 24 % during the CO2 flux measurements.
Mean July soil moisture between 10 am and 5 pm was 30 %. Note that not all flux plots had
continuous temperature and moisture loggers; this might thus explain some of the
differences between the means."*

*We think that the comment about rainfall events is very important. Unfortunately, we cannot
say how much the GHG fluxes change after rainfall because we do not have measurements
from the same plot before, during, and after rain. We will acknowledge this in the Discussion:
"Rainfall events might also increase soil moisture levels and activate processes related to
methanogenesis, photosynthesis and respiration as well as nitrogen cycling. While our soil
moisture predictions should capture these variations in soil wetness, we only made
measurements once per plot under clear conditions and do not have information about how
GHG fluxes might respond to rainfall events. We might thus underestimate some of the
instantaneous and longer-term changes in GHG fluxes during and after rain. "*

*And in the Supplement:*
*"Measurements were made under clear weather conditions but it also rained during the
study period. Rainfall can impact the soil moisture conditions and thus GHG fluxes. It rained
on 8 days during July 2018, and three of the days had heavier rain (>8 mm per day, FMI
2018). We made flux measurements during two of these days because during the
measurement time, the conditions were sunny. Nevertheless, the three heavier rainfall days
had clear but small impacts on soil moisture (volumetric water content (%) increased by
0.01-0.08) and it took approximately 0-3 days for soil moisture to return to the preceding soil
moisture level after the rain (Fig. X). Our data show that the range and mean of CH4 flux is
similar both in the plots measured during or 1-3 days after the rain and during other days,
suggesting that rainfall events did not have a major influence on our results. The mean CH4
flux during or 1-3 days after the rain was -1.8 and range -4.7 and 0.2 mg C m-2 d-1 (n=14),
and during other days -1.5 (mean), and -4.9 and 0.1 (range) mg C m-2 d-1 (n=72); note that
wetlands were not considered in this comparison because of their uneven distribution during
these time periods. In our upscaling framework, we control for the rainfall events as the GHG
flux predictions are based on bi-hourly soil moisture and temperature maps that should
reflect changes in soil moisture conditions after rain. "*

[Figure]

*Fig X. A figure showing the soil moisture variation during the study period from 5 example plots representing the vegetation types. The subplot shows how soil moisture changes after the rain. Vegetation types are b=barren, ds=deciduous shrub, es=evergreen shrub, g=meadow, w=wetland.*

Perhaps I missed it but - How many plots there were for each vegetation type?

 • Were samples sizes even between types? Weighted by spatial coverage?

*Response: The sample sizes can be found in Table S2. The sample sizes were not even between vegetation types, rather they represent the spatial coverage of each vegetation type. We will add a sentence about this to the main text.*

I am concerned by the use of regression forest methods a dataset of this size. With 10 inputs, but only 101 flux samples (no temporal), it seems to me these models are severely over parameterized. I doubt that there are sufficient training samples for the models to adequately parse out the functional relationships in 10D feature space. It might be beneficial to consider pruning your model - you could use the feature importance to inform your choice of which variables to keep/remove. This would likely result in a more robust model that is less likely to produce spurious results.

 • Random forest models are poorly suited for projection, often performing worse than simple linear regression (Hengl et al. 2018).

  – Hengl, T., Nussbaum, M., Wright, M., Heuvelink, G., & Graeler, B. (2018). Ran dom forest as a generic framework for predictive modeling of spatial and spatio temporal variables. PeerJ, 6, e5518. https://doi.org/10.7717/peerj.5518

 • Looking a the partial dependence plots, the support vector machine appears to produce more reasonable results and it would be nice to see some discussion of why.

• I would like to see the authors incorporate a simpler method like ordinary least squares regression to their ensemble. I would also like to see a breakdown of the upscaled estimates for each model, in addition to the estimates for the ensemble median.

*Response: In our GHG flux models, we had 101 samples and 8 predictors. While we have analyzed the model predictive performance in a rigorous and standard cross validation approach where the model parameters have also been tuned in a way that should mitigate overfitting, we understand the concern raised by the referee and agree that we are using complex machine learning models with a relatively small sample size. We will test how a generalized additive model, a simpler regression model, works in our modeling framework and compare our results with that. We will also calculate model fit, i.e., how well the models predict to the model training data to evaluate potential model overfitting. We will explore the possibility of removing predictors in our modeling framework. However, we feel that the fact that we have a similar set of predictors for all the GHG fluxes is also one of the key strengths of this study because it allows for a comparison of variable importances and response shapes.*

*Regarding the partial dependence plots, tree-based approaches often find thresholds in the data which are reflected as "wigglier" response shapes. This is related to the nature of the tree-based models as they split decision trees based on rules that can create these thresholds in the derived relationships. SVMs create smoother responses as the model is not based on decision trees; SVMs map the data into a high-dimensional space and build a hyperplane to separate the data and estimate smoother relationships. While we agree that very "wiggly" response shapes produced by the tree-based approaches that show no clear overall sign of positive/negative direction are highly uncertain and not suitable for large-scale extrapolations, SVMs also have their own strengths and limitations. For example, they might predict unrealistically high fluxes if the models need to extrapolate as the responses do not plateau in the same way as RFs and GBMs do. Thus, each model has its own strengths and limitations, and no model is perfect - therefore, it is generally recommended to use ensemble models in predictive efforts which we have done as well.*

*Taking a deeper look at the partial dependence plots, we want to highlight that most of the"wiggliest" partial dependence plots are found for variables that are less important or are produced by a model that has a low $R^2$ (e.g., the plot between DOC and N2O flux). To acknowledge this, we will change the y axis of all the plots to have a similar scale for each of the response variables as suggested by the referee later in the referee report. This way some of the wiggly response shapes with minor variable importance will likely only show a straight line in the partial dependence plot. We will add the following text to the Fig 5 caption so that the reader understands why the response shapes are different:*
*"RFs and GBMs are based on decision trees, where trees are split based on a certain threshold in the data, which can be seen as thresholds in the partial dependence plots as well. SVMs map the data into a high-dimensional space where a hyperplane is fit to separate them, creating smoother response shapes."*

*We will also add a figure to the Supplement showing the GHG flux predictions with all the three models.*

**Specific Comments**

**Introduction**

**Line 60:** "and they have different spatiotemporal dynamics with each other and compared to CO2 fluxes" - reads weird, consider rewording? "All three gasses have distinct spatiotemporal dynamics."

*Response: Thank you for this and all the other language suggestions below. They are extremely helpful.*

**Line 66:** Close parenthesis

*Response: We will close the parenthesis.*

**Materials and Methods**

**Line 93:** above *a* mountain birch

*Response: We will change this.*

**Line 105 - 106:** "101 GHG flux measurement plots and 50 to 5280 plots with other environ mental data" - this is confusing? 50 to 5280 plots? please explain better.

*Response: We will add more details to Table S1 and change this to:*

*"Our study design covered an area of ca. 3 x 1.5 km and consisted of 101 plots with GHG flux measurements and their supporting environmental data. To produce continuous maps of soil temperature, moisture, vegetation type, biomass, soil C/N, soil carbon stock, and dissolved organic carbon, we utilized an extended dataset where some of the variables were measured only from 50 plots while others were measured from close to 6000 plots (Table S1). "*

**Line 121:** Consider rephrasing - from Table S1 it looks like most factors had near complete coverage, so maybe say something like: "Environmental conditions explaining these GHG fluxes were measured at each plot. Most environmental variables had near complete spatial coverage; missing data were filled using the environmental predictions"

*Response: We will change this.*

**Line 168:** "Five gas samples were taken within a 50-min enclosure time" - this seems like a very long sampling interval? Are you concerned about heating within the chamber during the 50 min closure time disconnecting processes within the chamber from ambient conditions? Or about underestimating fluxes from high emitting wetland plots due to a reduction in the gas concentration gradient between the soil and the chamber head space? What was the rational for using this long sampling interval?

*Response: This is a good point. For CO2, we used a 90-second measurement time, so this potential issue only applies to N2O and CH4 fluxes. We will add the following text to the Supplement:*

*"We hypothesized most of the N2O fluxes and CH4 uptake fluxes to be small in this landscape dominated by upland tundra, and therefore used a 50-min chamber enclosure time to detect small changes in these concentrations (for a similar closure time, see e.g. (Marushchak et al. 2021; Voigt et al. 2017) . We used an opaque chamber, covered by space tape that reflects the sun, and did not thus observe any clear signs of heating of the chamber. The chamber headspace temperature difference during the start and end of the measurement ranged from -2.3 to 0.5 degrees (25th and 75th quantiles). Despite the long chamber enclosure time, the relationship between CH4 concentrations and measurement time at sites with high CH4 emissions (wetlands) was linear, indicating no issues with the chamber closure time (see Fig. X)."*

[Figure]

*Fig X. Example graphs showing the development of CH4 concentrations at some of the wetland sites.*

*Marushchak, M. E., J. Kerttula, K. Diáková, A. Faguet, J. Gil, G. Grosse, C. Knoblauch, et al. 2021. "Thawing Yedoma Permafrost Is a Neglected Nitrous Oxide Source." Nature Communications 12 (1): 7107.*
*Voigt, Carolina, Richard E. Lamprecht, Maija E. Marushchak, Saara E. Lind, Alexander Novakovskiy, Mika Aurela, Pertti J. Martikainen, and Christina Biasi. 2017. "Warming of Subarctic Tundra Increases Emissions of All Three Important Greenhouse Gases - Carbon Dioxide, Methane, and Nitrous Oxide." Global Change Biology 23 (8): 3121–38.*

**Line 207-208:** "We also utilized a larger dataset of 5820 vegetation descriptions from the

study design to create the vegetation type map. Please elaborate on how this data and how the map was created. Was this product created by a previous study? If so we need the citation. If not, you need give a more detailed description of the methods and data used to create the map.

*Response: We will edit the sentence:*

*"We utilized a larger dataset of 5820 vegetation descriptions estimated in the field and from aerial imagery from the study design to create the vegetation type map (for more details, see S4.1)."*

**Line 239:** I don't think one sentence necessitates its own sub-section. Perhaps expand on this a bit or merge it with section 2.2.3

*Response: We will edit this.*

**Lines 266-268:** I feel this is insufficient justification for the choice of models. I would like to see a bit more on why these methods were chosen and the pros/cons of each.

*Response: We will add the following text to the Methods:*

*"These three approaches are non-parameteric and can handle linear and non-linear relationships. We chose RFs and GBMs because they utilize several decision trees in an ensemble model framework and thus avoid overfitting, have high accuracy, are highly adaptable, and are not significantly impacted by outliers. We chose SVMs because they are good at generalizing the relationships in the data."*

**Line 302:** Seems like a reasonable thing to do - I assume the idea is to minimize the effect of outliers on the model? Perhaps explicitly say that in the text?

*Response: The idea is to normalize the distribution of errors, which is one of the assumptions in regression analyses. Non-normal distribution of residuals can bias, for example, the calculation of prediction intervals.*

**Line 320:** Leave one out is concerning?

*Response: Leave-one-out approach is a widely used cross-validation approach. We are not fully certain what the referee means with this comment.*

**Results**

**Lines 356-358:** "The scatter plots of observed and predicted GHG fluxes suggest that the highest flux estimates are often predicted most poorly, but the mean fluxes in each vegetation type were predicted accurately."

- This seems like an obvious point - empirical models will tend toward the mean of the training domain regardless of how well fit the distribution of the individual training points.

*Response: We agree and will mention that this was expected.*

**Line 371:** net *uptake* of CO2

*Response: We will change this.*

**Discussion**

**416-417:** "Aboveground plant biomass and vegetation type were important drivers for both which suggests a dominance of autotrophic (plant) respiration over heterotrophic (microbial) respiration." - this statement seems like a bit of a stretch? I would assume more above ground biomass also means more litter input for decomposition, and also would likely be correlated with below ground biomass » leading to greater microbial decomposition of root exudates? How strongly correlated were the input parameters?

*Response: We agree with this and will remove the sentence.*

*Biomass and SOC have a correlation of -0.14 (p=0.16). The correlation is negative because the largest biomass with high Betula nana or Empetrum cover are often found in drier soils with low soil carbon stocks, whereas the wetlands have the highest soil carbon stocks but small-moderate vegetation biomass. We will add this and the other correlations to a new Supplementary table.*

**Lines 459-461:** Out of curiosity, for what portion of the year do you expect these favorable conditions to last? I'd imagine some of the sinks, especially the valley bottom meadow would be sources during snow-melt period, and possibly again during the freeze up period in fall?

*Response: This is an interesting question. Unfortunately we do not have any data covering the entire period from snow melt to thaw in this study design, but we did a CO2 flux measurement campaign in 2019 where we sampled a smaller study area three times during the snow-free season. Those data suggest that during the early growing season (mid-late June), the ecosystems were on average CO2 neutral (data published only in a Finnish-language Master's thesis; https://helda.helsinki.fi/handle/10138/331463). We agree with the referee that the meadows are likely CO2 sources straight after snow melt or right before snow arrival. This is because during the spring there is an inflow of carbon and other nutrients from meltwater streams that likely boost decomposition and during the autumn deciduous leaves of graminoids have senesced and only soil respiration is active. Interestingly though, the thesis suggested that across the smaller study design, Reco decreased more than GPP towards late summer; however, we did not capture the freeze up period in our sampling. A new year-round eddy covariance tower will be set up in this landscape in 2024 which will provide more insight on the temporal dynamics of this ecosystem.*

**Line 503:** The gasses themselves do not act as sinks/sources, consider rephrasing.

*Response: We will change this.*

**Line 510-511:** "evergreen or deciduous shrub expansion may increase or decrease the growing season GHG sink" - consider switching to "deciduous or evergreen shrub expansion may …" to better get your point across.

*Response: We will change this.*

**Line 536:** perhaps a bit of a stretch to say "all the main vegetation types"?

*Response: We will say "most of the main vegetation types" instead.*

**Line 543:** would the temporal variability of soil moisture and temperature contribute to the difference too? What was the variability like over the period - relative to the sample days? It would be nice to se a time series of soil temperature and moisture (averaged by vegetation type) over the July 2018 study period - with the sample dates/times highlighted for reference.

*Response: We controlled for the variability in soil moisture and temperature over the study period in our upscaling approach, thus this has been considered in these average flux comparisons. We will make sure that the new manuscript has time series graphs of soil moisture and temperature.*

**Line 556:** or models having too many parameters …

*Response: We will mention model structures too.*

**Line 575-577:** An important point, well put!

*Response: Thank you!*

**Figures**

**Figure 1:** Plots a. and c. color schemes are misleading - using the same colors to show very different phenomena. I suggest changing the color for the vegetation maps to one better suited for discrete qualitative data. For plot b. the chambers should be color-coded by vegetation type so the reader can better see the spatial distribution of each type. Additionally, it would be helpful if the figure caption (or somewhere else in the text) said how many chambers there were for each vegetation type. For the numeric data in c. soil temperature and annual soil temperature are the colormaps are inverted relative to the other plots, which also makes things a bit unclear at first glance.

*Response: We have picked one color scheme that we use throughout the text that has colors that are clearly distinguishable from each other and can easily be read by color-blind readers (or black and white paper) as well. We have inverted the color scales of temperature maps so that blue reflects cold conditions. We will test if changing the color scheme for the vegetation type makes the figure easier to read. We will add the number of chambers for each vegetation type to the figure caption. We will also color the points with the vegetation type information. We will also clarify what the vegetation type legends refer to, as this was not clear for referee 1.*

**Figure 3:** Could you make the boxplot larger so they're easier to read and consider excluding the points that are within the boxes - makes for a confusing/overly complex boxplot. Additionally:
- You could set the y-axis limits for GPP and ER to the same values for a more direct comparison.
- You have a "-0" on your boxplot for biomass
- Maybe just keep the most important plots and move some to the appendix to save

space?

*Response: We will remove the variables that were not used in the model (pH, nitrogen stock) and make the boxplots bigger so that they are easier to read. We will also set the y-axis limits for GPP and ER to the same values and change "-0" to "0".*

**Figure 5:** Are these importance values normalized to a 0-1 scale?

• Should the bars sum to 1 for each model?
• Why is the importance for N2O so low for SVM across all features
• Please use a common y-axis range across the subplots for easier comparison
• Any estimate of uncertainty in these feature importance estimates? e.g., for a RF model you can get the importance of each sub-model and use it to calculate a 95% CI over around the RF feature importance values.

*Response: The bars have been calculated using the permutation approach with the idea that if we randomly permute the values of an important predictor in the training data, the training performance would degrade (since permuting the values of a predictor destroys relationships between that predictor and the response variable). We did not normalize them between 0 and 1 and the bars should not sum to 1 but each bar is theoretically limited between 0 and 1 because the approach compares R^2 values.*

*We will add the following text to the main text:*

*"The importance for variables explaining the N2O flux is low because the model predictive performance is close to random. Variable importance scores were calculated using a permutation approach with the idea that if we randomly permute the values of an important predictor in the training data, the training performance would degrade. However, for N2O fluxes this random permutation had minimal effect on the predictors."*

*We used 100 simulations to calculate 100 importance scores which were eventually averaged, but we will show the variability in those importance scores too. The differences in importance scores across the models also provides information about uncertainty.*

**Figure 6:** It seems to me this plot highlights how regression trees (RF and GBM) models are poorly suited for this type of analysis, particularly given the small sample size. The noisy response curves they generate are because regression trees are **treating each terminal node in a tree as a discrete data point to match rather than a continuous response function to fit.** The idea behind a random forest, is that averaging many of these over-fit trees will emulate the desired response function (*only within the bounds of the training data*). However, the there do not appear to be sufficient samples for the RF model to be able to do that.

• Are these partial dependence yhat values in the same unit/scale for each predictor? If so it would be useful to have each y-axis on the same scale to see the relative magnitude of the partial dependence by variable.

*Response: We are not sure if one can conclude from these graphs that the response curves for RF and GBM are noisy. After all, most of the highly important variables in models that performed well have relatively clear response shapes with some minor wiggliness. The thresholds that exist in the partial dependence plots are simply related to the nature of tree-based approaches as well as thresholds that make ecological sense. For example, the sharp jump between the soil moisture and CH4 flux data can be seen by a bivariate plot as well, because with soil moisture at around 60 % soils become saturated which boosts methanogenesis (see figure below). We will change the y axis scale of the partial dependence plots, this is a great point. For a longer response to this, see our response earlier to the major points raised by this referee.*

[Figure]

Figure. The relationship between observed instantaneous soil moisture and CH4 flux. Units: volumetric water content (%) and mg CH4 m$^{-2}$ d$^{-1}$.

**Figure 7:** Using "strong" and "moderate" to describe the CH4 sink strength of upland areas seems odd. By area, yes they may be large sinks overall, but on a per unit area basis they are not given their relatively small magnitude compared to wetland CH4 emissions shown in Fig 8. Perhaps try rephrasing the labels in the images? Alternatively, show your landscape map here to emphasize that you're talking on a "per landscape fraction" basis.

*Response: We will change the language here.*

**Figure 8:** I would like to see a different color for the error bars to help them stand out from the plots more.

*Response: We will use a different color (e.g., grey) so that they can be better separated from the blue bars.*

**Supplement:**

**Figure S4:** Needs a legend for the plots.

*Response: We will add the legend.*

---

## Author Response (AR1)

**Overall referee comments**

We made substantial edits to the manuscript. In short, we redid all the modeling and uncertainty analysis and redraw all the figures to improve the manuscript in terms of issues with transformed variables, model overfitting, and some visuals. We also added the missing details related to methods, and text describing the uncertainties of our study. The key conclusions and results remained mostly the same, except the entire landscape was a small CH4 source instead of a small CH4 sink.

**Referee comments 1 RC1**

This study provides excellent insight into spatial heterogeneity of GHG fluxes in a fairly unprecedented sampling extent of chamber-based fluxes. The authors compensate well for the lack of temporal coverage in fluxes by leveraging the spatial information available through a hierarchical Bayesian modeling approach to NEE. The remote sensing analysis is thorough. The comparison of scaling models provides useful insight into important multivariate influences on GHG fluxes.

*Response: Thank you for this feedback!*

I have two areas of significant improvement to address. The first is a lack of information provided about the Bayesian hierarchical modeling. The authors reference Williams et al. 2006 for their model structure. This should be explicitly provided, along with the parameters that are being estimated, at least in the supplement. The authors mention using vague prior information for these parameters. The prior distributions and initial values used for the MCMC chains should be provided (at least in the supplement). There is no mention of posterior predictive checks or tests for convergence. These are necessary to ensure the model is appropriate for the data and that parameters are estimated correctly (without need for a longer burn-in for example). There should be some presentation of the posterior distributions for parameter estimates. There could be some additional discussion as well related to how much the random effect of plot contributed predictions, or how variable the random effect was within vegetation types, etc.

*Response: Thank you for pointing this out. We have clarified that ER, maximum GPP, half-saturation constant, and an exponential air temperature response of ER were the parameters in the main text.*

[Figure]

*Supporting figure for revision. The distribution of NEE observations (y) and draws from the posterior predictive distribution (yrep) in micromoles $CO_2$ $m^{-2}$ $s^{-1}$.*

*We added the following details to the Supplement:*

*"We used weakly informative priors for the plot-specific intercept terms, informed by those reported in Williams et al. 2006 and Happonen et al 2022. The means and standard deviations for the priors were 1 and 2 for the logarithm of the ER (posterior mean: 0.65 and posterior standard deviation 0.13), 0 and 1 for the temperature effect on logarithm of the ER (posteriors: 0.02 and 0.01), 10 and 10 for maximum GPP (posteriors: 8.54 and 0.52), and 6.2 and 0.3 for the logarithm of the half-saturation parameter (posteriors: 5.92 and 0.06). We used logarithms for some parameters to normalize their error distribution. For more details on the model structure, see Happonen et al. 2022 (section 2.4.1) and the code light_response_model.R in Virkkala et al. 2023.*

*The Bayesian R2 of the model was 0.96, all family- and population-specific mean parameters had an Rhat less than 1.03, and posterior predictive draws matched well with the observations, indicating model convergence and good predictive performance. "*

The second area for improvement is related to using back-transformations of log-transformed predictions. From my understanding, soil C, biomass, and soil moisture were log-transformed during their upscaling. Then they were back-transformed and subsequently used as drivers to predict GHG fluxes. Back transforming a prediction (from a non-affine transformation) will introduce bias that needs to be corrected. For a useful explanation of the problem, see this blogpost: https://florianwilhelm.info/2020/05/honey_i_shrunk_the_target_variable/. There are multiple methods available for correcting back-transformation bias, some of which are analytical such as in the case of simple linear regression. See this paper for a comparison of several bias

correction methods for GBM models: https://kdd-milets.github.io/milets2022/papers/MILETS_2022_paper_0925.pdf. Since all three of the back-transformed variables rank as fairly high predictors, and are especially important at high soil C, high soil moisture, etc where the back-transformation bias is larger, it is critical to correct this bias. The CH4 flux scaling similarly needs a back transformation bias correction, since a cube-root transformation is also non-affine, and these predicted fluxes are subsequently back transformed for comparing to in situ fluxes and calculating carbon budgets.

*Response: We agree that back-transformations are problematic because they impact both the error distribution and the shape of the regression. While this can certainly be an issue for the environmental datasets (soil C, biomass, and soil moisture), we think that the largest uncertainties in our study were with the back-transformation of the CH4 flux variable as it was one of our main GHG flux response variables. We analyzed how these transformations impact the distribution of upscaled environmental conditions and fluxes and discovered that there were indeed severe biases. For example, average in-situ wetland CH4 emissions were more than four times higher than average wetland CH4 emissions from the upscaling with back-transformed CH4 flux data.*

*We tried to correct for this bias using the simple linear regression regression approach as suggested by the referee. However, while this improved model performance at the highest fluxes, it caused issues with low CH4 fluxes. Consequently, low CH4 fluxes were overestimated. See figures visualizing observed and predicted fluxes (using the full model training data, i.e. not CV-based) below.*

[Figure]

*Supporting figure for revision. The scatterplots with observed (x axis) and predicted (y axis) fluxes of the ensemble model with the back-transformed CH4 flux (left), and bias-corrected back-transformed CH4 flux (right) with a 1:1 line showing that fluxes below 100 mg C d-1 would be overestimated and small fluxes <0 mg C d-1 would be underestimated with the bias-correction approach. This is not a desired result as it biases the most frequent flux magnitudes in the landscape.*

*Because of this issue, we decided to proceed without the bias-correction and continued exploring how the model performs without any transformations. For CH4, with the slightly edited version of the model (one predictor less now; see comments to referee 2) the scatterplots with observed and predicted fluxes were not much different when compared with the model with transformed variables (CH4 flux, soil moisture). We thus decided to remove all the transformations from the models. This decision was backed up by the notion that machine learning models should be relatively flexible with the data distributions and should not need transformations in the first place. For example, tree-based models are based on creating splits on the data where the potential non-normal distribution of the data should not have a big effect on the splits. Below is the figure of the CH4 flux model with and without transformed variables.*

[Figure]

*Supporting figure for revision. Scatterplots with observed (y axis) and predicted (x axis) CH4 fluxes with the transformed and then back-transformed CH4 flux (left) and non-transformed CH4 flux (right) with a 1:1 line showing that there are no major differences between the transformed and non-transformed models.*

*Nevertheless, this was an important aspect to correct in the revised version, because using non-transformed data changed the landscape-level CH4 budget from a small sink to a small source, suggesting that we were earlier underestimating CH4 emissions from wetlands. The average CH4 fluxes estimated with upscaling are now also closer to the in-situ CH4 flux averages. Sentences related to these findings were corrected throughout the text.*

Minor comments:

There are numerous regressions demonstrating model performance (Fig 4, FigS3), with the r-squared reported. The slopes intercepts, and p-values should also be reported, as this would help assess performance and bias in the model predictions.

*Response: After careful consideration, we decided not to include these statistics to the manuscript. This decision was made based on the fact that there is no common consensus on how the slopes should be calculated in terms of whether the observed or predicted values should be treated as a response in this calculation, and this might create confusion when interpreting the statistics. Many studies often have the predicted values as a response variable even though it is recommended to have the observed values as a response variable (see https://www.sciencedirect.com/science/article/abs/pii/S0304380008002305).*

Font sizes for Fig 3 are too small.

*Response: We reorganized the figure and removed some less important boxplots to make the fonts bigger.*

It is unclear what the 'Agency 2017' reference is in Fig 1. It is also unclear what the colored vegetation boxes correspond to in panel (c) of Fig 1.

*Response: We corrected and clarified these.*

**Referee comments 2**

**General Comments**

The authors use flux observations from 101 chamber plots and a dataset of environmental drivers to estimate daytime CO2, CH4 and N2O fluxes across a study site in northern Finland over July 2018. This dataset is unique and the topic is certainly within the scope of the Biogeosciences. I think the manuscript could make a valuable contribution to the literature on high latitude greenhouse gas fluxes. However, I feel the article requires major revisions to address some key pitfalls before it can be published.

While there are a large number of sample plots, the lack of temporal replicates concerns me. I understand the difficulty and expense associated with Arctic research, so I do not feel this issue alone should disqualify the manuscript, but this limitation should be discussed more in depth. The authors briefly address this but I feel it necessitates further explanation within the text itself.

*Response: Thank you for this feedback. We agree that in an ideal world, we would have study designs covering both the fine-scale spatial heterogeneity as well as the temporal variability, complemented by more controlled experiments to verify the drivers of change. In this study, however, we were primarily focusing on the spatial variation, as we had identified the lack of extensive spatial study settings from the literature. Doing the GHG flux chamber measurements alone took one month for 3 researchers, and unfortunately we did not have the resources to continue this throughout several months. We added sentences to the main text emphasizing that spatial study designs can be used to infer correlations between*

*variables, but correlation does not imply causation. We also discussed if and how the relationships that we observe across spatial gradients compare with those observed in time-series studies where causal relationships can be more easily observed. See e.g. line 555:*

*"...the relationships we observed were logical and comparable to those observed in other studies - both based on spatial and time series study designs (e.g., positive soil moisture-CH4 flux or soil temperature-ER relationships (Euskirchen et al. 2014; Davidson et al. 2016; Zona et al. 2023)). Moreover, our study is based on a dataset focusing on spatial variation in GHG fluxes and correlations between variables. Therefore, the dataset should not directly be used to infer causal relationships or estimates of flux change over time (Damgaard 2019), and we advise caution when extrapolating these results to areas outside our study domain or different time periods."*

- Sampling spanned only two days, between 10 am to 5 pm. Given this - I have a hard time believing "the spatial variation in our plots covered most of the temperature variation during the growing season" without a more thorough discussion.

  – Sec S1 gives mean air & soil temperature for the chambers during observation and over the study period. I would like to see these broken down in more detail, **with soil moisture too**. Perhaps as a boxplot in the supplement?
  – I assume these samples were collected under clear weather conditions. Fluxes during and after any rainfall events would be quite different. Was there much rain in July 2018? Perhaps rainfall days should be excluded from upscaling? Is it possible an upland site that is otherwise a sink could shift to a CH4 source during/after rainfall?

*Response: Thank you for pointing out the importance of the temporal representativeness of the data. The sampling spanned a month, from July 1st to August 2nd. The snow melts in May-June and plants reach their maximum biomass in July-early August, after which the autumn and senescence slowly start. We assume that clarifying this misunderstanding likely solves part of the issues raised by the referee in this comment.*

*We added the following description to the Supplement:*
*"Mean soil moisture was 27 % during the CH4 and N2O flux measurements and 24 % during the CO2 flux measurements. Mean July soil moisture between 10 am and 5 pm was 30 %. Note that not all flux plots had continuous temperature and moisture loggers; the partly different distribution of microclimate loggers might thus explain some of the differences between the means."*

*We think that the comment about rainfall events is very important. Unfortunately, we cannot say how much the GHG fluxes change after rainfall because we do not have measurements from the same plot before, during, and after rain. We acknowledged this in the Discussion on line 588:*

*"Rainfall events are another source of uncertainty in our upscaling because they might also increase soil moisture levels and activate processes related to methanogenesis, photosynthesis and respiration as well as nitrogen cycling. While our soil moisture*

*predictions should capture these variations in soil wetness, we only made measurements once per plot under clear conditions and do not have information about how GHG fluxes might respond to rainfall events. We might thus underestimate some of the instantaneous and longer-term changes in GHG fluxes during and after rain (see Text S1 and Fig. S10 for details)."*

*And in the Supplement:*
*"Measurements were made under clear weather conditions but it also rained during the study period. Rainfall can impact the soil moisture conditions and thus GHG fluxes. It rained on 8 days during July 2018, and three of the days had heavier rain (>8 mm per day, FMI 2018). We made flux measurements during two of these days because during the measurement time, the conditions were sunny. Nevertheless, the three heavier rainfall days had clear but small impacts on soil moisture (volumetric water content (%) increased by 0.01-0.08 units) and it took approximately 0-3 days for soil moisture to return to the preceding soil moisture level after the rain (Fig. S10). Our data show that the range and mean of CH4 flux is similar both in the plots measured during or 1-3 days after the rain and during other days, suggesting that rainfall events did not have a major influence on our results. The mean CH4 flux during or 1-3 days after the rain was -1.8, and range -4.7 and 0.2 mg C m-2 d-1 (n=14), and during other days -1.5 (mean), and -4.9 and 0.1 (range) mg C m-2 d-1 (n=72); note that wetlands were not considered in this comparison because of their uneven distribution during these time periods which also impacts the summary statistics due to their high VWC% levels. In our upscaling framework, we control for the rainfall events as the GHG flux predictions are based on bi-hourly soil moisture and temperature maps that should reflect changes in soil moisture conditions after rain. "*

[Figure]

*Fig. S10. A figure showing the soil moisture variation during the study period from 5 example plots representing the vegetation types. The subplot shows how soil moisture changes after*

*the rain. Vegetation types are b=barren, ds=deciduous shrub, es=evergreen shrub, g=meadow, w=wetland.*

Perhaps I missed it but - How many plots there were for each vegetation type?

• Were samples sizes even between types? Weighted by spatial coverage?

*Response: The sample sizes can be found in Table S2. The sample sizes were not even between vegetation types, rather they roughly represent the spatial coverage of each vegetation type. We added a sentence about this to the main text.*

I am concerned by the use of regression forest methods a dataset of this size. With 10 inputs, but only 101 flux samples (no temporal), it seems to me these models are severely over parameterized. I doubt that there are sufficient training samples for the models to adequately parse out the functional relationships in 10D feature space. It might be beneficial to consider pruning your model - you could use the feature importance to inform your choice of which variables to keep/remove. This would likely result in a more robust model that is less likely to produce spurious results.

• Random forest models are poorly suited for projection, often performing worse than simple linear regression (Hengl et al. 2018).

– Hengl, T., Nussbaum, M., Wright, M., Heuvelink, G., & Graeler, B. (2018). Ran dom forest as a generic framework for predictive modeling of spatial and spatio temporal variables. PeerJ, 6, e5518. https://doi.org/10.7717/peerj.5518

• Looking a the partial dependence plots, the support vector machine appears to produce more reasonable results and it would be nice to see some discussion of why.

• I would like to see the authors incorporate a simpler method like ordinary least squares regression to their ensemble. I would also like to see a breakdown of the upscaled estimates for each model, in addition to the estimates for the ensemble median.

*Response: In our GHG flux models, we originally had 101 samples and 8 predictors. While we have analyzed the model predictive performance in a rigorous and standard cross validation approach where the model parameters have also been tuned in a way that should mitigate overfitting, we understand the concern raised by the referee and agree that we are using complex machine learning models with a relatively small sample size. Consequently, we made several new analyses to address this comment. We summarized all of these in the supplement under S5.3 Diagnosing model overfitting and added language related to the associated uncertainties in the main text. Supplementary table S3 also summarizes the performance of individual models for comparison.*

*We calculated the model fit statistic which showed high $R^2$ (0.33 to 0.94 excluding N2O fluxes) and low error values which were clearly better compared to the cross-validated estimates (e.g., $R^2$ 0.23 to 0.53 excluding N2O fluxes). This demonstrated that our models were potentially overfitting. We added sentences about this on line 550:*

*"However, at the same time, our models showed some signs of overfitting as demonstrated by the high model fit statistics and the mismatch between model fit and predictive*

*performance statistics (Supplementary Text S5.3). This is a common issue in upscaling (Kemppinen et al. 2018; Shi et al. 2022), and could indicate that the models have potentially learned to fit some noise or specific patterns unique to the training set instead of broadly generalizable relationships. Nevertheless, the relationships we observed were logical and comparable to those observed in other studies - both based on spatial and time series study designs (e.g. positive soil moisture-CH4 flux or soil temperature-ER relationships (Euskirchen et al. 2014; Davidson et al. 2016; Zona et al. 2023)).”*

*As removing predictors generally helps with model overfitting, we removed one predictor variable that was not important for any of the response variables: dissolved organic carbon. Other variables were important for at least some of the response variables, and we wanted to keep the same predictor variables in each model because we considered it as one of the key strengths of this study because it allows for a comparison of variable importances and response shapes. However, we also tested if model overfitting can be decreased by including only 3 predictors. This decreased model predictive performance and model fit but only slightly (e.g., ca. 0.01-0.1 units for $R^2$); thus, the model fit was still quite high compared to the predictive performance estimates, and there was still a mismatch between model fit and predictive performance estimates. This result did not convince us to make machine learning models with less predictors.*

*We also tested generalized additive models (GAMs). In GAMs, the difference between model fit and predictive performance estimates was slightly smaller than in machine learning models, making this a promising approach to deal with model overfitting, although GAM model performance was overall lower compared to machine learning estimates (see Supplementary Table S3). We used gaussian distribution for all the fluxes and smooth responses for the continuous variables, and qqplots showed no issues with the residuals. In the predictions, GAM extrapolated flux values that were unrealistic, with CH4 flux sinks being predicted up to -33 mg C m-2 d-1 (minimum in-situ flux -4.9), and with wetlands showing both large emissions and strong sinks right next to each other, even after testing with different family distributions and combinations of predictors. Thus, the GAM-based upscaled fluxes were much more uncertain compared to the machine learning-based fluxes, and because of this it was excluded from the final analyses. In the future, studies should explore if regression models developed using the Bayesian framework would be more suitable for flux upscaling because they allow the user to control, for example, the prior distributions for parameters in an improved way. As a conclusion, we kept the same machine learning models but removed one predictor, and added text about model overfitting issues to the text.*

*Regarding the partial dependence plots, tree-based approaches often find thresholds in the data which are reflected as “wigglier” response shapes. This is related to the nature of the tree-based models as they split decision trees based on rules that can create these thresholds in the derived relationships. SVMs create smoother responses as the model is not based on decision trees; SVMs map the data into a high-dimensional space and build a hyperplane to separate the data and estimate smoother relationships. While we agree that very “wiggly” response shapes produced by the tree-based approaches that show no clear overall sign of positive/negative direction are highly uncertain and not suitable for large-scale extrapolations, SVMs also have their own strengths and limitations. For example, they might predict unrealistically high fluxes if the models need to extrapolate as the responses do not plateau in the same way as RFs and GBMs do. Thus, each model has its own strengths and*

*limitations, and no model is perfect - therefore, it is generally recommended to use ensemble models in predictive efforts which we have done as well.*

*Taking a closer look at the partial dependence plots, we want to highlight that most of the "wiggliest" partial dependence plots are found for variables that are less important or are produced by a model that has a low $R^2$ (e.g., the plot between DOC and N2O flux). To acknowledge this, we changed the y axis of all the plots to have a similar scale for each of the response variables as suggested by the referee later in the referee report. This way some of the wiggly response shapes with minor variable importance only showed a relatively stable straight line in the partial dependence plot. We added the following text to the Fig 5 caption so that the reader understands why the response shapes are different:*
*"RFs and GBMs are based on decision trees, where trees are split based on a certain threshold in the data, which can be seen as thresholds in the partial dependence plots as well. SVMs map the data into a high-dimensional space where a hyperplane is fit to separate them, creating smoother response shapes."*

**Specific Comments**

**Introduction**

**Line 60:** "and they have different spatiotemporal dynamics with each other and compared to CO2 fluxes" - reads weird, consider rewording? "All three gasses have distinct spatiotemporal dynamics."

*Response: Thank you for this and all the other language suggestions below. They are extremely helpful.*

**Line 66:** Close parenthesis

*Response: We closed the parenthesis.*

**Materials and Methods**

**Line 93:** above *a* mountain birch

*Response: We changed this.*

**Line 105 - 106:** "101 GHG flux measurement plots and 50 to 5280 plots with other environ mental data" - this is confusing? 50 to 5280 plots? please explain better.

*Response: We added more details to Table S1 and changed this to:*

*"Our study design covered an area of ca. 3 x 1.5 km and consisted of 101 plots with GHG flux measurements and their supporting environmental data (Fig. 1). To produce continuous maps of soil temperature, moisture, vegetation type, biomass, soil C/N, and soil organic carbon stock, we utilized an extended dataset where some of the variables were measured*

*from 50 plots while others from close to 6000 plots (Table S1)"*

**Line 121:** Consider rephrasing - from Table S1 it looks like most factors had near complete coverage, so maybe say something like: "Environmental conditions explaining these GHG fluxes were measured at each plot. Most environmental variables had near complete spatial coverage; missing data were filled using the environmental predictions"

*Response: We changed this.*

**Line 168:** "Five gas samples were taken within a 50-min enclosure time" - this seems like a very long sampling interval? Are you concerned about heating within the chamber during the 50 min closure time disconnecting processes within the chamber from ambient conditions? Or about underestimating fluxes from high emitting wetland plots due to a reduction in the gas concentration gradient between the soil and the chamber head space? What was the rational for using this long sampling interval?

*Response: This is a good point. For CO2, we used a 90-second measurement time, so this potential issue only applies to N2O and CH4 fluxes. We added the following text to the Supplement:*

*"We hypothesized that most of the N2O fluxes and CH4 uptake fluxes to be small in this landscape dominated by upland tundra, and therefore used a 50-min chamber enclosure time to detect small changes in these concentrations (for a similar closure time, see e.g. (Marushchak et al. 2021; Voigt et al. 2017)) . We used an opaque chamber, covered by space tape that reflects the sun, and did not thus observe any clear signs of heating of the chamber. The chamber headspace temperature difference during the start and end of the measurement ranged from -2.3 to 0.5 ° C (25th and 75th quantiles). Despite the long chamber enclosure time, the relationship between CH4 concentrations and measurement time at sites with high CH4 emissions (wetlands) was linear, indicating no issues with the chamber closure time (see Fig. S9)."*

*Plot 12223:*                  *Plot 12207:*

[Figure]

*Plot 12215:*                                    *Plot 12209:*

[Figure]

*Fig. S9. Example graphs showing the development of CH4 concentrations at some of the wetland sites.*

**Line 207-208:** "We also utilized a larger dataset of 5820 vegetation descriptions from the study design to create the vegetation type map. Please elaborate on how this data and how the map was created. Was this product created by a previous study? If so we need the citation. If not, you need give a more detailed description of the methods and data used to create the map.

*Response: We edited the sentence:*

*"We utilized a larger dataset of 5820 vegetation descriptions estimated in the field and from aerial imagery from the study design to create the vegetation type map (for more details, see S4.1)."*

**Line 239:** I don't think one sentence necessitates its own sub-section. Perhaps expand on this a bit or merge it with section 2.2.3

*Response: Done*

**Lines 266-268:** I feel this is insufficient justification for the choice of models. I would like to see a bit more on why these methods were chosen and the pros/cons of each.

*Response: We will add the following text to the Methods:*

*"These three approaches are non-parameteric and can handle linear and non-linear relationships and different data distributions. We chose RFs and GBMs because they utilize several decision trees in an ensemble model framework and thus avoid overfitting, have high accuracy, are highly adaptable, and are not significantly impacted by outliers. We chose SVMs because they are good at generalizing the relationships in the data."*

**Line 302:** Seems like a reasonable thing to do - I assume the idea is to minimize the effect of outliers on the model? Perhaps explicitly say that in the text?

*Response: Based on the other referee's comment, this was now edited. See the response earlier.*

**Line 320:** Leave one out is concerning?

*Response: Leave-one-out approach is a widely used cross-validation approach. We are not fully certain what the referee means with this comment.*

**Results**

**Lines 356-358:** "The scatter plots of observed and predicted GHG fluxes suggest that the highest flux estimates are often predicted most poorly, but the mean fluxes in each vegetation type were predicted accurately."

- This seems like an obvious point - empirical models will tend toward the mean of the training domain regardless of how well fit the distribution of the individual training points.

*Response: We agree and will mention that this was expected.*

**Line 371:** net *uptake* of CO2

*Response: We changed this.*

**Discussion**

**416-417:** "Aboveground plant biomass and vegetation type were important drivers for both which suggests a dominance of autotrophic (plant) respiration over heterotrophic (microbial) respiration." - this statement seems like a bit of a stretch? I would assume more above ground biomass also means more litter input for decomposition, and also would likely be correlated with below ground biomass » leading to greater microbial decomposition of root exudates? How strongly correlated were the input parameters?

*Response: We agree with this and removed the sentence.*

*Biomass and SOC have a correlation of -0.14 (p=0.16). The correlation is negative because the largest biomass with high Betula nana or Empetrum cover are often found in drier soils with low soil carbon stocks, whereas the wetlands have the highest soil carbon stocks but small-moderate vegetation biomass.*

**Lines 459-461:** Out of curiosity, for what portion of the year do you expect these favorable conditions to last? I'd imagine some of the sinks, especially the valley bottom meadow would be sources during snow-melt period, and possibly again during the freeze up period in fall?

*Response: This is an interesting question. Unfortunately we do not have any data covering the entire period from snow melt to thaw in this study design, but we did a CO2 flux measurement campaign in 2019 where we sampled a smaller study area three times during the snow-free season. Those data suggest that during the early growing season (mid-late June), the ecosystems were on average CO2 neutral (data published only in a Finnish-language Master's thesis; https://helda.helsinki.fi/handle/10138/331463). We agree with the referee that the meadows are likely CO2 sources straight after snow melt or right before snow arrival. This is because during the spring there is an inflow of carbon and other nutrients from meltwater streams that likely boost decomposition and during the autumn deciduous leaves of graminoids have senesced and only soil respiration is active. Interestingly though, the thesis suggested that across the smaller study design, Reco decreased more than GPP towards late summer; however, we did not capture the freeze up period in our sampling. A new year-round eddy covariance tower will be set up in this*

*landscape in 2024 which will provide more insight on the temporal dynamics of this ecosystem.*

**Line 503:** The gasses themselves do not act as sinks/sources, consider rephrasing.

*Response: We changed this.*

**Line 510-511:** "evergreen or deciduous shrub expansion may increase or decrease the growing season GHG sink" - consider switching to "deciduous or evergreen shrub expansion may …" to better get your point across.

*Response: We changed this.*

**Line 536:** perhaps a bit of a stretch to say "all the main vegetation types"?

*Response: We had the word "almost" before.*

**Line 543:** would the temporal variability of soil moisture and temperature contribute to the difference too? What was the variability like over the period - relative to the sample days? It would be nice to se a time series of soil temperature and moisture (averaged by vegetation type) over the July 2018 study period - with the sample dates/times highlighted for reference.

*Response: We controlled for the variability in soil moisture and temperature over the study period in our upscaling approach, thus this has been considered in these average flux comparisons. See our responses earlier.*

**Line 556:** or models having too many parameters …

*Response: We mentioned model structures too.*

**Line 575-577:** An important point, well put!

*Response: Thank you!*

**Figures**

**Figure 1:** Plots a. and c. color schemes are misleading - using the same colors to show very different phenomena. I suggest changing the color for the vegetation maps to one better suited for discrete qualitative data. For plot b. the chambers should be color-coded by vegetation type so the reader can better see the spatial distribution of each type. Additionally, it would be helpful if the figure caption (or somewhere else in the text) said how many chambers there were for each vegetation type. For the numeric data in c. soil temperature and annual soil temperature are the colormaps are inverted relative to the other plots, which also makes things a bit unclear at first glance.

*Response: We have picked one color scheme that we use throughout the text that has colors that are clearly distinguishable from each other and can easily be read by color-blind readers (or black and white paper) as well. We have inverted the color scales of temperature maps so that blue reflects cold conditions. We have slightly reorganized the figure and hope it is more clear now in terms of vegetation type maps. We did not add the vegetation type plot details because we wanted to have an equal amount of emphasis on all the environmental conditions in this figure.*

**Figure 3:** Could you make the boxplot larger so they're easier to read and consider excluding the points that are within the boxes - makes for a confusing/overly complex boxplot. Additionally:
- You could set the y-axis limits for GPP and ER to the same values for a more direct comparison.

- You have a "-0" on your boxplot for biomass
- Maybe just keep the most important plots and move some to the appendix to save

space?

*Response: We removed the variables that were not used in the model (pH, dissolved organic carbon, nitrogen stock) and made the boxplots bigger so that they are easier to read. We set the y-axis limits for GPP and ER to the same values and changed "-0" to "0".*

**Figure 5:** Are these importance values normalized to a 0-1 scale?

- Should the bars sum to 1 for each model?
- Why is the importance for N2O so low for SVM across all features
- Please use a common y-axis range across the subplots for easier comparison
- Any estimate of uncertainty in these feature importance estimates? e.g., for a RF model you can get the importance of each sub-model and use it to calculate a 95% CI over around the RF feature importance values.

*Response: The bars have been calculated using the permutation approach with the idea that if we randomly permute the values of an important predictor in the training data, the training performance would degrade (since permuting the values of a predictor destroys relationships between that predictor and the response variable). We did not normalize them between 0 and 1 and the bars should not sum to 1 but each bar is theoretically limited between 0 and 1 because the approach compares $R^2$ values.*

*We added the following text to the main text on line 281:*

*"We used 100 simulations to calculate 100 importance scores which were averaged. A standard deviation across these scores was used as an uncertainty estimate, together with the differences in average importance across models. "*

*The importance for variables explaining the N2O flux is low because the model predictive performance is close to random, therefore the permutation also has minimal impact on the model performance.*

**Figure 6:** It seems to me this plot highlights how regression trees (RF and GBM) models are poorly suited for this type of analysis, particularly given the small sample size. The noisy response curves they generate are because regression trees are **treating each terminal node in a tree as a discrete data point to match rather than a continuous response function to fit.** The idea behind a random forest, is that averaging many of these over-fit trees will emulate the desired response function (*only within the bounds of the training data*). However, the there do not appear to be sufficient samples for the RF model to be able to do that.

- Are these partial dependence yhat values in the same unit/scale for each predictor? If so it would be useful to have each y-axis on the same scale to see the relative magnitude of the partial dependence by variable.

*Response: We politely disagree with the assessment that  the response curves for RF and GBM are noisy. After all, most of the highly important variables in models that performed well have relatively clear response shapes with some minor wiggliness. The thresholds that exist in the partial dependence plots are simply related to the nature of tree-based approaches as well as thresholds that make ecological sense. For example, the sharp jump between the soil*

*moisture and CH4 flux data can be seen by a bivariate plot as well, because with soil moisture at around 60 % soils become water-saturated which boosts methanogenesis (see figure below). However, based on this comment, we changed the y axis scale of the partial dependence plots, which was a great suggestion. For a longer response to this, see our response earlier to the major points raised by this referee.*

[Figure]

*Supporting figure for revision. The relationship between observed instantaneous soil moisture and CH4 flux. Units: volumetric water content (%) and mg CH4 $m^{-2}$ $d^{-1}$.*

**Figure 7:** Using "strong" and "moderate" to describe the CH4 sink strength of upland areas seems odd. By area, yes they may be large sinks overall, but on a per unit area basis they are not given their relatively small magnitude compared to wetland CH4 emissions shown in Fig 8. Perhaps try rephrasing the labels in the images? Alternatively, show your landscape map here to emphasize that you're talking on a "per landscape fraction" basis.

*Response: We changed the language here.*

**Figure 8:** I would like to see a different color for the error bars to help them stand out from the plots more.

*Response: We used a different color (grey) so that they can be better separated from the blue bars.*

**Supplement:**

**Figure S4:** Needs a legend for the plots.

*Response: We added the legend.*

---

## Referee Report (RR1)

**Re-review of "High-resolution spatial patterns and drivers of terrestrial ecosystem carbon dioxide, methane, and nitrous oxide fluxes in the tundra"**

**General Comments**

I thank the authors' for their responses to my numerous comments and they have done a thorough job addressing my concerns. I particularly appreciate the change in scale of the partial dependence plots, as it makes it much easier to the relationships (or lack thereof) with drivers and compare between drivers. I also appreciate the additional discussion of over-fitting and the work to address it by removing an dissolved oxygen. I don't have any further comments to add and I look forward to seeing the published manuscript.